Corrected: Publisher correction

# A general and flexible method for signal extraction from single-cell RNA-seq data

Davide Risso [1], Fanny Perraudeau[2], Svetlana Gribkova[3], Sandrine Dudoit[2,4] & Jean-Philippe Vert [5,6,7,8]

Single-cell RNA-sequencing (scRNA-seq) is a powerful high-throughput technique that enables researchers to measure genome-wide transcription levels at the resolution of single cells. Because of the low amount of RNA present in a single cell, some genes may fail to be detected even though they are expressed; these genes are usually referred to as dropouts. Here, we present a general and flexible zero-inflated negative binomial model (ZINB-WaVE), which leads to low-dimensional representations of the data that account for zero inflation (dropouts), over-dispersion, and the count nature of the data. We demonstrate, with simulated and real data, that the model and its associated estimation procedure are able to give a more stable and accurate low-dimensional representation of the data than principal component analysis (PCA) and zero-inflated factor analysis (ZIFA), without the need for a preliminary normalization step.

[1] Division of Biostatistics and Epidemiology, Department of Healthcare Policy and Research, Weill Cornell Medicine, New York, NY 10065, USA. [2] Division of Biostatistics, School of Public Health, University of California, Berkeley, CA 94720, USA. [3] Laboratoire de Probabilités et Modèles Aléatoires, Université Paris Diderot, 75005 Paris, France. [4] Department of Statistics, University of California, Berkeley, CA 94720, USA. [5] CBIO-Centre for Computational Biology, MINES ParisTech, PSL Research University, 75006 Paris, France. [6] Institut Curie, 75005 Paris, France. [7] INSERM U900, 75005 Paris, France. [8] Department of Mathematics and Applications, Ecole Normale Supérieure, 75005 Paris, France. Correspondence and requests for materials should be addressed to S.D. (email: sandrine@stat.berkeley.edu) or to J.-P.V. (email: jean-philippe.vert@ens.fr)

Single-cell RNA-sequencing (scRNA-seq) is a powerful and relatively young technique enabling the characterization of the molecular states of individual cells through their transcriptional profiles[1]. It represents a major advance with respect to standard "bulk" RNA-sequencing, which is only capable of measuring average gene expression levels within a cell population. Such averaged gene expression profiles may be enough to characterize the global state of a tissue, but completely mask signal coming from individual cells, ignoring tissue heterogeneity. Assessing cell-to-cell variability in expression is crucial for disentangling complex heterogeneous tissues[2–4] and for understanding dynamic biological processes, such as embryo development[5] and cancer[6]. Despite the early successes of scRNA-seq, to fully exploit the potential of this new technology, it is essential to develop statistical and computational methods specifically designed for the unique challenges of this type of data[7].

Because of the tiny amount of RNA present in a single cell, the input material needs to go through many rounds of amplification before being sequenced. This results in strong amplification bias, as well as dropouts, i.e., genes that fail to be detected even though they are expressed in the sample[8]. The inclusion in the library preparation of unique molecular identifiers (UMIs) reduces amplification bias[9], but does not remove dropout events, nor the need for data normalization[10,11]. In addition to the host of unwanted technical effects that affect bulk RNA-seq, scRNA-seq data exhibit much higher variability between technical replicates, even for genes with medium or high levels of expression[12].

The large majority of published scRNA-seq analyses include a dimensionality reduction step. This achieves a two-fold objective: (i) the data become more tractable, both from a statistical (cf. curse of dimensionality) and computational point of view; (ii) noise can be reduced while preserving the often intrinsically low-dimensional signal of interest. Dimensionality reduction is used in the literature as a preliminary step prior to clustering[3,13,14], the inference of developmental trajectories[15–18], spatio-temporal ordering of the cells[5,19], and, of course, as a visualization tool[20,21]. Hence, the choice of dimensionality reduction technique is a critical step in the data analysis process.

A natural choice for dimensionality reduction is principal component analysis (PCA), which projects the observations onto the space defined by linear combinations of the original variables with successively maximal variance. However, several authors have reported on shortcomings of PCA for scRNA-seq data. In particular, for real data sets, the first or second principal components often depend more on the proportion of detected genes per cell (i.e., genes with at least one read) than on an actual biological signal[22,23]. In addition to PCA, dimensionality reduction techniques used in the analysis of scRNA-seq data include independent components analysis (ICA)[15], Laplacian eigenmaps[18,24], and t-distributed stochastic neighbor embedding (t-SNE)[2,4,25]. Note that none of these techniques can account for dropouts, nor for the count nature of the data. Typically, researchers transform the data using the logarithm of the (possibly normalized) read counts, adding an offset to avoid taking the log of zero.

Recently, Pierson & Yau[26] proposed a zero-inflated factor analysis (ZIFA) model to account for the presence of dropouts in the dimensionality reduction step. Although the method accounts for the zero inflation typically observed in scRNA-seq data, the proposed model does not take into account the count nature of the data. In addition, the model makes a strong assumption regarding the dependence of the probability of detection on the mean expression level, modeling it as an exponential decay. The fit on real data sets is not always good and, overall, the model lacks flexibility, with its inability to include covariates and/or normalization factors.

Here, we propose a general and flexible method that uses a zero-inflated negative binomial (ZINB) model to extract low-dimensional signal from the data, accounting for zero inflation (dropouts), over-dispersion, and the count nature of the data. We call this approach Zero-Inflated Negative Binomial-based Wanted Variation Extraction (ZINB-WaVE). The proposed model includes a sample-level intercept, which serves as a global-scaling normalization factor, and gives the user the ability to include both gene-level and sample-level covariates. The inclusion of observed and unobserved sample-level covariates enables normalization for complex, non-linear effects (often referred to as batch effects), whereas gene-level covariates may be used to adjust for sequence composition effects, such as gene length and GC-content effects. ZINB-WaVE is an extension of the RUV model[27,28], which accounts for zero inflation and over-dispersion and for which unobserved sample-level covariates may either capture variation of interest or unwanted variation. We demonstrate, with simulated and real data, that the model and its associated estimation procedure are able to give a more stable and accurate low-dimensional representation of the data than PCA and ZIFA, without the need for a preliminary normalization step. The approach is implemented in the open-source R package *zinbwave*, publicly available through the Bioconductor Project (https://bioconductor.org/packages/zinbwave).

## Results

**ZINB-WaVE is a general and flexible model for scRNA-seq.** ZINB-WaVE is a general and flexible model for the analysis of high-dimensional zero-inflated count data, such as those recorded in single-cell RNA-seq assays. Given $n$ samples (typically, $n$ single cells) and $J$ features (typically, $J$ genes) that can be counted for each sample, we denote with $Y_{ij}$ the count of feature $j$ ($j = 1,…, J$) for sample $i$ ($i = 1, …, n$). To account for various technical and biological effects, typical of single-cell sequencing technologies, we model $Y_{ij}$ as a random variable following a ZINB distribution (see Methods for details).

Both the mean expression level ($\mu$) and the probability of dropouts ($\pi$) are modeled in terms of observed sample-level and gene-level covariates ($X$ and $V$, respectively, Fig. 1). In addition, we include a set of unobserved sample-level covariates ($W$) that need to be inferred from the data. The matrix $X$ can include covariates that induce variation of interest, such as cell types, or covariates that induce unwanted variation, such as batch or quality control (QC) measures. It can also include a constant column of ones for an intercept that accounts for gene-specific differences in mean expression level or dropout rate (cf. scaling in PCA). The matrix $V$ can also accommodate an intercept to account for cell-specific global effects, such as size factors representing differences in library sizes (i.e., total number of reads per sample). In addition, $V$ can include gene-level covariates, such as gene length or GC-content.

The unobserved matrix $W$ contains unknown sample-level covariates, which could correspond to unwanted variation as in RUV[27,28] or could be of interest as in cluster analysis (e.g., cell type). The model extends the RUV framework to the ZINB distribution (thus far, RUV had only been implemented for linear[27] and log-linear regression[28]). It differs in interpretation from RUV in the $W\alpha$ term, which is not necessarily considered unwanted; this term generally refers to unknown low-dimensional variation, that could be due to unwanted technical effects (as in RUV), such as batch effects, or to biological effects of interest, such as cell cycle or cell differentiation.

It is important to note that although $W$ is the same, the matrices $X$ and $V$ could differ in the modeling of $\mu$ and $\pi$, if we assume that some known factors do not affect both. When $X = \mathbf{1}_n$

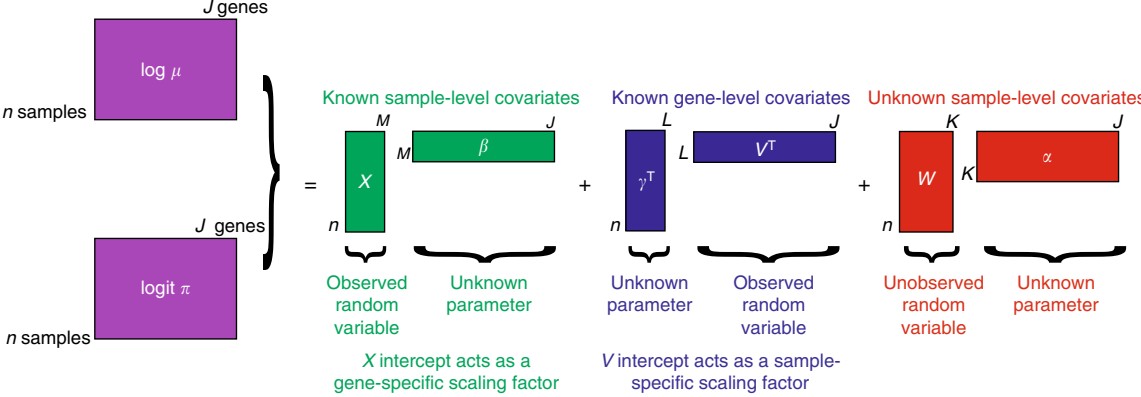

**Fig. 1** Schematic view of the ZINB-WaVE model. Given $n$ cells and $J$ genes, let $Y_{ij}$ denote the count of gene $j$ ($j = 1,..., J$) for cell $i$ ($i = 1,..., n$) and $Z_{ij}$ an unobserved indicator variable, equal to one if gene $j$ is a dropout in cell $i$ and zero otherwise. Then, $\mu_{ij} = E[Y_{ij}|Z_{ij} = 0, X, V, W]$ and $\pi_{ij} = \Pr(Z_{ij} = 1|X, V, W)$. We model $\ln(\mu)$ and $\text{logit}(\pi)$ with the regression specified in the figure. Note that the model allows for different covariates to be specified in the two regressions; we have omitted the $\mu$ and $\pi$ indices for clarity (see Methods for details)

and $V = \mathbf{1}_J$, the model is a factor model akin to PCA, where $W$ is a factor matrix and $\alpha_\mu$ and $\alpha_\pi$ are loading matrices. However, the model is more general, allowing the inclusion of additional sample and gene-level covariates that might help to infer the unknown factors.

**ZINB-WaVE leads to biologically meaningful clusters**. We applied the ZINB-WaVE procedure to several publicly available real data sets, from microfluidics, plate-based, and droplet-based platforms (see Methods).

As previously shown[22,23], the first few principal components of scRNA-seq data, even after normalization, can be influenced by technical rather than biological features, such as the proportion of genes with at least one read (detection rate) or the total number of reads (sequencing depth) per sample. Figure 2 illustrates this using the V1 data set: although the first two principal components somewhat segregated the data by layer of origin (Fig. 2a), the clustering was far from perfect. This is at least partly due to unwanted technical effects, such as sequencing depth and amount of starting material. To quantify such technical effects, we computed a set of QC measures, such as detection rate and total number of reads (see Methods). The first two principal components are especially correlated with detection rate (Fig. 2b).

ZIFA suffered from the same problem: the clustering of the samples in two dimensions was not qualitatively different from PCA (Fig. 2c) and the second component was highly correlated with detection rates (Fig. 2d).

Conversely, ZINB-WaVE led to tighter clusters, grouping the cells by layer of origin (Fig. 2e). Furthermore, the two components inferred by ZINB-WaVE showed lower correlation with the QC features (Fig. 2f), highlighting that the clusters shown in Fig. 2e are not driven by technical effects.

We repeated these analyses on the S1/CA1 data set (Supplementary Fig. 1), mESC data set (Supplementary Fig. 2), and glioblastoma data set (Supplementary Fig. 3). For all data sets, ZINB-WaVE led to tighter clusters in two dimensions. However, it did not always lead to a decrease in correlation with the QC measures. See also Hicks et al.[22] for additional data sets in which principal components are strongly correlated with detection rate.

As a measure of the goodness of the clustering results, we used the average silhouette width (see Methods), computed using the labels available from the original study: these were either known a priori (e.g., the patient ID in the glioblastoma data set) or inferred from the data (and validated) by the authors of the original

publication (e.g., the cell types in the S1/CA1 data set). For all four data sets, ZINB-WaVE led to generally tighter clusters, as shown by an increased per-cluster average silhouette width in the majority of the groups (Supplementary Fig. 5).

**ZINB-WaVE leads to novel biological insights**. To demonstrate the ability of ZINB-WaVE to lead to novel biological insights, we focused on two inferential questions typical of scRNA-seq studies: (i) the identification of developmental lineages and (ii) the characterization of rare cell types.

First, we re-analyzed a set of cells from the mouse olfactory epithelium (OE) that were collected to identify the developmental trajectories that generate olfactory neurons (mOSN), sustentacular cells (mSUS), and microvillous cells (MV)[29]. In the original publication, the data were normalized by full-quantile normalization, followed by a regression-based adjustment for sample quality. Clustering on the first 50 principal components identified cellular states that were then ordered into developmental lineages by Slingshot[30] using the first five PCs. In the original analysis, Slingshot was able to correctly infer the lineages in its supervised mode, by manually specifying lineage endpoints (i.e., clusters corresponding to the mature cell types). Figure 3a shows the minimum spanning tree (MST) obtained with the described supervised analysis. When running Slingshot in unsupervised mode, however, the inferred MST only correctly identified the neuronal (mOSN) and microvillous (MV) lineages, while it was unable to identify sustentacular (mSUS) cells as a mature cell type (Fig. 3b). By repeating the clustering and lineage reconstruction with Slingshot on the first 50 factors of ZINB-WaVE, we were able to infer the correct lineages even in unsupervised mode (Fig. 3c). This suggests that the low-dimensional signal revealed by ZINB-WaVE more closely matches the true developmental process. See Perraudeau et al.[31] for a complete workflow that involves ZINB-WaVE for dimensionality reduction and Slingshot for lineage reconstruction.

We next focused on a set of 68,579 peripheral blood mononuclear cells (PBMCs) assayed with the 10× Genomics Chromium system[32]. This example allowed us to demonstrate how ZINB-WaVE can be applied to a state-of-the-art data set that comprises tens of thousands of cells, while also illustrating that existing clustering algorithms can be used on the low-rank matrix inferred by ZINB-WaVE. In particular, we applied a popular clustering method, based on the identification of shared nearest neighbors[33], similar to that of Macosko et al.[2] and implemented

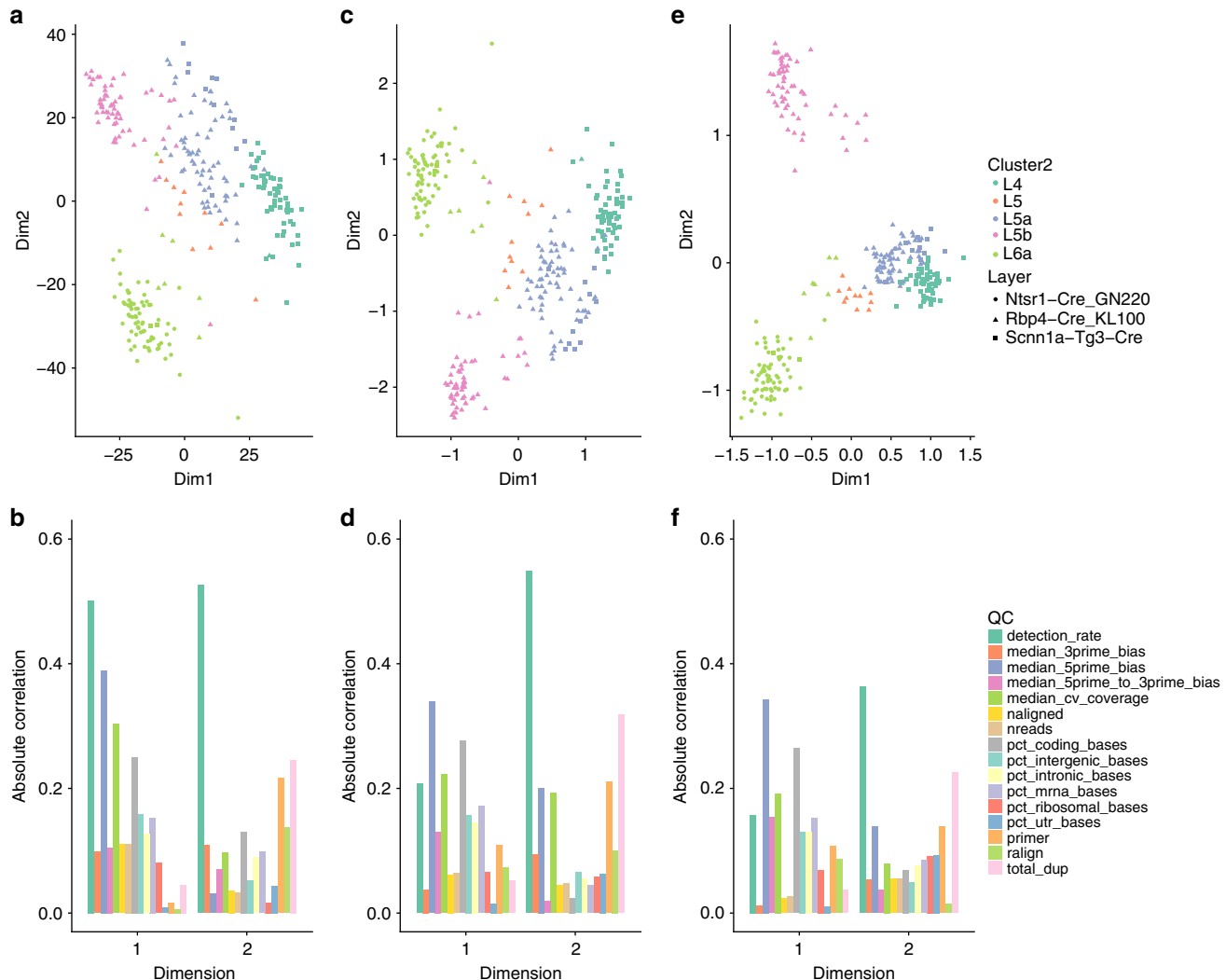

**Fig. 2** Low-dimensional representation of the V1 data set. Upper panels provide two-dimensional representations of the data, after selecting the 1000 most variable genes. Lower panels provide barplots of the absolute correlation between the first two components and a set of QC measures (see Methods). **a**, **b** PCA (on TC-normalized counts). **c**, **d** ZIFA (on TC-normalized counts). **e**, **f** ZINB-WaVE (no normalization needed). ZINB-WaVE leads to a low-dimensional representation that is less influenced by technical variation and to tighter, biologically meaningful clusters. The ZINB-WaVE projection was robust to the number of genes selected (Supplementary Fig. 4)

in the R package Seurat[34]. We modified the clustering procedure to use ZINB-WaVE (with $K = 10$) instead of PCA as the dimensionality reduction step. To visualize the clustering results, we applied t-SNE to the inferred $W$ matrix (Fig. 3d).

Our approach recapitulates the major cell populations found in Zheng et al.[32] (Fig. 3d, e). In particular, 80% of the cells are T-cells (Clusters 0–1 CD4+ T-cells; Clusters 2–6 CD8+ T-cells). As in Zheng et al.[32], we are able to identify populations of activated cytotoxic T-cells (Cluster 5; 9%), natural killer cells (Cluster 7; 6%), and B-cells (Cluster 8; 5%) (Fig. 3d). In addition, we are able to identify sub-populations of myeloid cells that were not completely characterized in the original publication. In particular, we identified clusters corresponding to: CD14+ monocytes (Cluster 9; 3%), characterized by the expression of *CD14*, *S100A9*, *LYZ*[35]; CD16+ monocytes (Cluster 10; 2.5%), which express *FCGR3A/CD16*, *AIF1*, *FTL*, *LST1*[35]; CD1C+ dendritic cells (DC; Cluster 11; 1%), which express *CD1C*, *LYZ*, *HLA-DR*[35]; plasmacytoid dendritic cells (pDC; Cluster 13; 0.5%), expressing *GZMB*, *SERPINF1*, *ITM2C*[35]; megakaryocyte (Cluster 15; 0.2%), expressing *PF4*[32] (Fig. 3e). We also identified several small clusters (collectively comprising about 1% of the cells) that were

either enriched for mitochondrial genes (Cluster 14) or ribosomal genes (Clusters 17 and 18), or for which we could not find any markers (Clusters 12 and 16). We hypothesize that these clusters represent either doublets or lower quality libraries.

The above analysis differs from that of the original publication not only in terms of dimensionality reduction, but also in terms of clustering (Zheng et al.[32] used *k*-means on the first 50 PCs). We hence repeated the clustering using a sequential procedure based on *k*-means (see Methods for details). This led to similar (albeit noisier) results (Supplementary Fig. 6), highlighting that ZINB-WaVE is able to extract meaningful biological signal from the data, as input to a variety clustering procedures. In fact, extracting a two-dimensional signal from ZINB-WaVE already allowed us to identify most major cell types (Supplementary Fig. 7).

**Impact of normalization**. As for any high-throughput genomic technology, an important aspect of scRNA-seq data analysis is normalization. Indeed, there are a variety of steps in scRNA-seq experiments that can introduce bias in the data and whose effects need to be corrected for[11]. Some of these effects, e.g., sequencing depth, can be captured by a global-scaling factor (usually referred

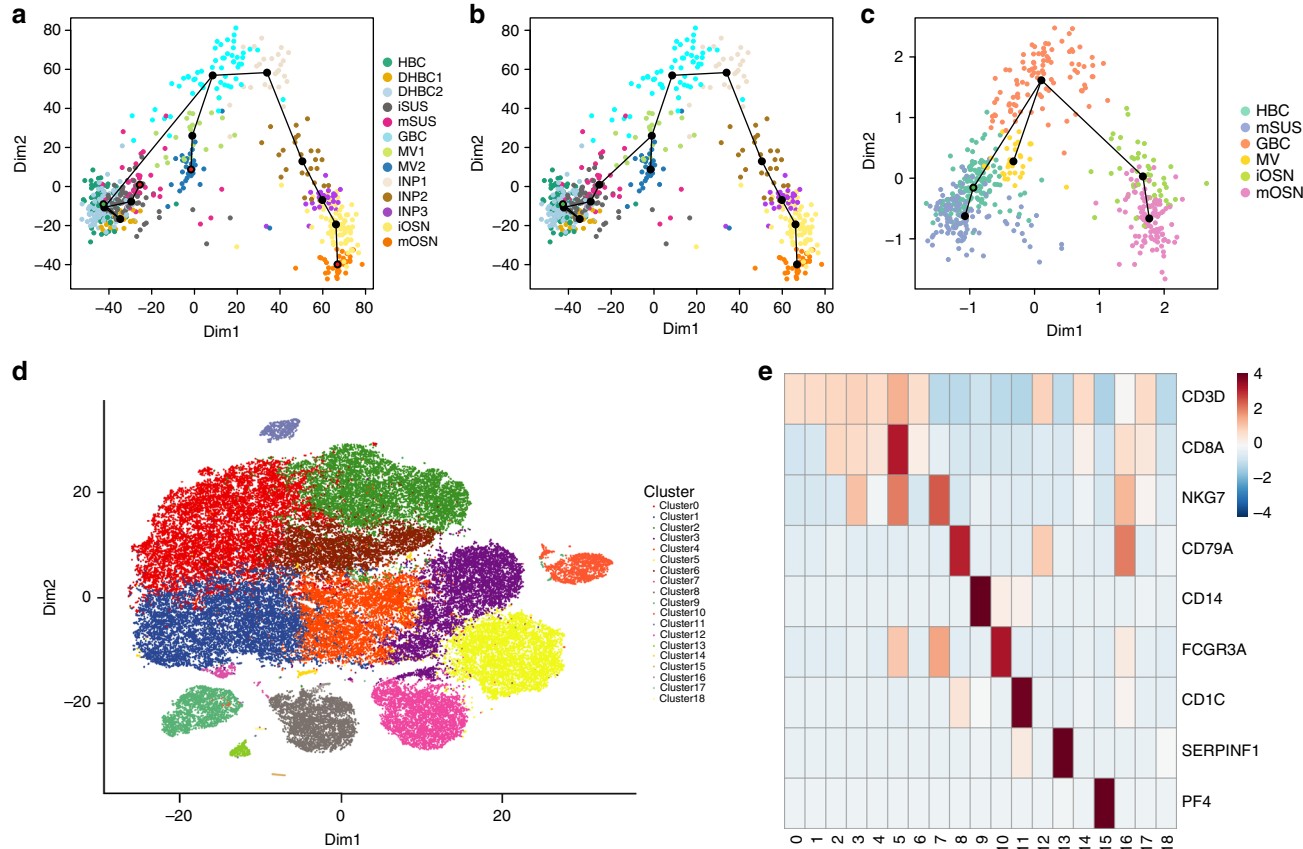

**Fig. 3** Using ZINB-WaVE to gain novel biological insights. **a–c** Lineage inference on the OE data set. Slingshot minimum spanning tree on cell clusters: **a** PCA with endpoint supervision (marked as red nodes in the tree). **b** PCA with no endpoint supervision. **c** ZINB-WaVE with no endpoint supervision. **a**, **b** RSEC clustering (see Methods) on the first 50 PCs of the normalized counts led to 13 clusters; **c** the same procedure on 50 components of W from ZINB-WaVE led to six clusters: horizontal basal cells (HBC); tran- sitional HBC (DHBC1-2); immature sustentacular cells (iSUS); mature sustentacular cells (mSUS); globose basal cells (GBC); microvillous cells (MV); immediate neuronal precursors (INP1-3); immature olfactory neurons (iOSN); mature olfactory neurons (mOSN). mOSN, MV, and mSUS are the mature cell types and should be identified as the three lineage endpoints. **d**, **e** Discovery of rare cell types for the 10× Genomics 68k PBMCs data set. **d** Scatterplot of first two t-SNE dimensions obtained from 10 components of W from ZINB-WaVE; cells are color-coded by cluster. Clustering was performed on the 10 components of W (see Methods). **e** Heatmap of expression measures for marker genes for the 18 clusters found by our procedure: columns correspond to clusters and rows to genes; the value in each cell is the average log expression measure per cluster, centered and scaled so that each row has mean zero and standard deviation one

to as size factor). Other more complex, non-linear effects, such as those collectively known as batch effects, require more sophisti- cated normalization[28]. Accordingly, a typical scRNA-seq pipeline will include a normalization step, i.e., a linear or non-linear transformation of read counts, to make the distributions of expression measures comparable between samples. The normal- ization step is usually carried out prior to any inferential proce- dure (e.g., clustering, differential expression analysis, or pseudotime ordering). In this work, we considered three popular between-sample normalization methods: total count (TC), trim- med mean of M values (TMM), and full-quantile (FQ) normal- ization (see Methods); we also compared the results to unnormalized data (RAW). TC, along with the related methods transcripts per million (TPM) and fragments per kilobase million (FPKM), is by far the most widely used normalization in the scRNA-seq literature; hence, TC-normalized data were used for the results shown in Fig. 2 and in Supplementary Figs. 1–3.

Normalization highly affected the projection of the data in lower dimensions (Supplementary Figs. 8–15) and, consequently, the clustering results varied greatly between normalization methods in all four data sets (Fig. 4). Strikingly, the choice of normalization method was more critical than the choice between PCA and ZIFA. For instance, for the mESC data set (Fig. 4d), FQ

normalization followed by either PCA or ZIFA led to very high average silhouette width. Critically, the ranking of normalization methods was not consistent across data sets (Fig. 4), highlighting that the identification of the best normalization method for a given data set is a difficult problem for scRNA-seq[11,36].

One important feature of the ZINB-WaVE model is that the sample-level intercept $\gamma$ (corresponding to a column of ones in the gene-level covariate matrix $V$, see Methods) acts as a global-scaling normalization factor, making a preliminary global-scaling step unnecessary. As a consequence, ZINB-WaVE can be directly applied to unnormalized read counts, hence preserving the distributional properties of count data. ZINB-WaVE applied to unnormalized counts led to results that were comparable, in terms of average silhouette width, to PCA and ZIFA applied using the best performing normalization (Fig. 4). In particular, ZINB-WaVE outperformed PCA and ZIFA on the S1/CA1 (Fig. 4b) and glioblastoma (Fig. 4c) data sets, while it was slightly worse than PCA (after FQ normalization) on the mESC data set (Fig. 4d) and than ZIFA (after FQ normalization) on the V1 data set (Fig. 4a). Interestingly, the overall average silhouette width was lower for the S1/CA1 and glioblastoma data sets than it was for the V1 and mESC data sets, suggesting that ZINB-WaVE leads to better clustering in more complex situations (e.g., lower sequencing depth).

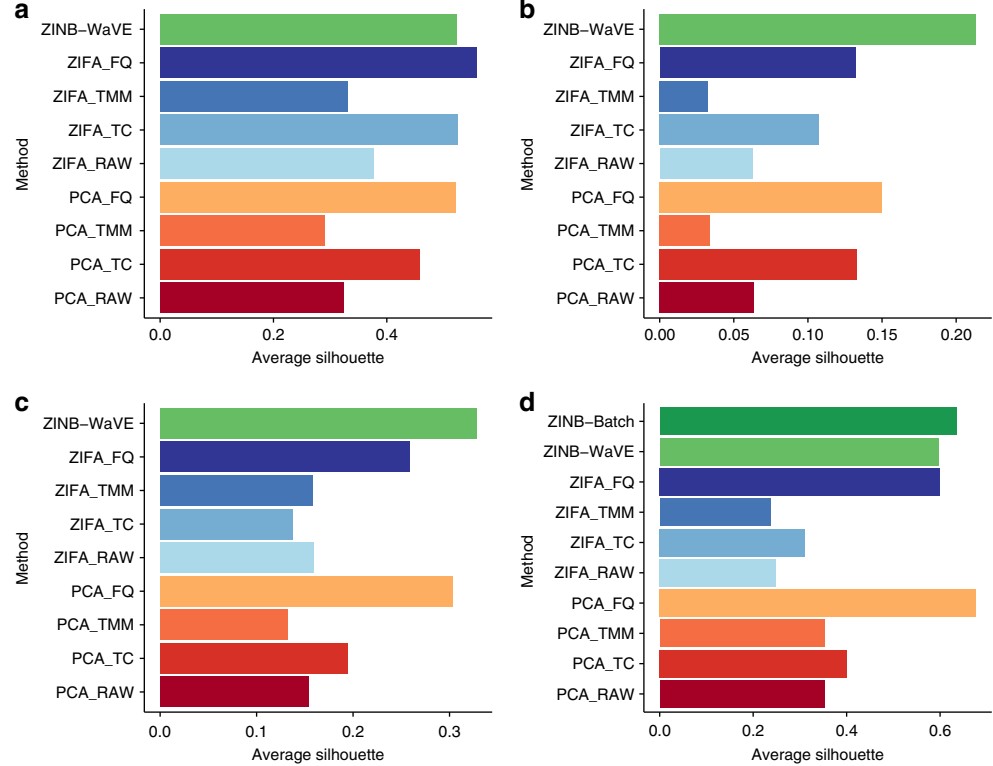

**Fig. 4** Average silhouette width in real data sets. **a** V1 data set; **b** S1/CA1 data set; **c** glioblastoma data set; **d** mESC data set. Silhouette widths were computed in the low-dimensional space, using the groupings provided by the authors of the original publications: unsupervised clustering procedure (**a**, **b**), observed characteristics of the samples, such as patient (**c**) and culture condition (**d**). PCA and ZIFA were applied after normalization: unnormalized (RAW), total count (TC), trimmed mean of M values (TMM), and full-quantile (FQ). For the mESC data set (**d**), we fitted the ZINB-WaVE model with batch as a sample-level covariate (ZINB-Batch) in addition to the default model with only a sample-level intercept (see Fig. 5)

Although the overall average silhouette width is a useful metric to rank the different methods in terms of how well they represent known global biological structure, looking only at the average across many different cell types may be misleading. For the V1 data set, for instance, ZINB-WaVE led to a slightly lower overall average silhouette width than ZIFA (Fig. 4a). However, ZINB-WaVE yielded higher cluster-level average silhouette widths for the L6a, L5b, and L5 samples, whereas ZIFA produced higher average silhouette widths for the L4 and L5a samples (Supplementary Fig. 5a). In fact, certain cell types may be easier to cluster than others, leading to silhouette widths that differ greatly between different clusters, as is the case for the S1/CA1 data set: oligodendrocytes are much easier to cluster than the other cell types and all methods were able to identify them (Supplementary Fig. 5b); on the other hand, only ZINB-WaVE was able to achieve a positive silhouette width for the pyramidal SS and CA1 neurons and it performed much better than PCA and ZIFA in the interneuron cluster; finally, certain groups, such as the endothelial-mural and astrocytes, were simply too hard to distinguish in three dimensions (Supplementary Fig. 5b).

**Accounting for batch effects**. Although the sample-level intercept $\gamma$ (corresponding to a column of ones in the gene-level covariate matrix $V$) can act as a global-scaling normalization factor, this may not be sufficient to accurately account for complex, non-linear technical effects that may influence the data (e.g., batch effects). Hence, we explored the possibility of including additional sample-level covariates in $X$ to account for such effects (see Methods).

We illustrate this using the mESC data set, which is a subset of the data described in Kolodziejczyk et al.[37] and which comprises

two batches of mESCs grown in three different media (see Methods). ZINB-WaVE applied with no additional covariates led to three clusters, corresponding to the three media (Fig. 5a; Supplementary Fig. 2). However, the clusters corresponding to media 2i and a2i are further segregated by batch (Fig. 5a). Including indicator variables for batch in ZINB-WaVE (as columns of $X$) removed such batch effects, consolidating the clustering by medium (Fig. 5b). This led to slightly larger average silhouette widths, both overall (Fig. 4d) and at the cluster level (Fig. 5c). Conversely, the average silhouette width for batch, a measure of how well the cells cluster by batch, was much larger for the model that did not include the batch covariate (Fig. 5d), indicating that explicitly accounting for batch in the ZINB-WaVE model effectively removed the dependence of the inferred low-dimensional space on batch effects. Similar results were obtained by normalizing the data with the ComBat batch correction method[38], implemented in the Bioconductor R package sva[39] (Supplementary Fig. 16). The advantage of ZINB-WaVE is the ability of including batch effects in the same model used for dimensionality reduction, without the need for prior data normalization.

The mESC data set is an example of good experimental design, where each batch includes cells from each biological condition (this is known as a factorial design). Hence, it is relatively easy to correct for batch effects and, unsurprisingly, both ComBat and ZINB-WaVE successfully do so. The glioblastoma data set is an example of a more complex situation, in which there is confounding between batch and biology, each patient being processed separately[22]. Luckily, the glioblastoma design is not completely confounded, as patient *MGH26* was processed in two batches (Fig. 5e). We used this feature of the data to test whether ZINB-WaVE was able to adjust for batch effects even in the

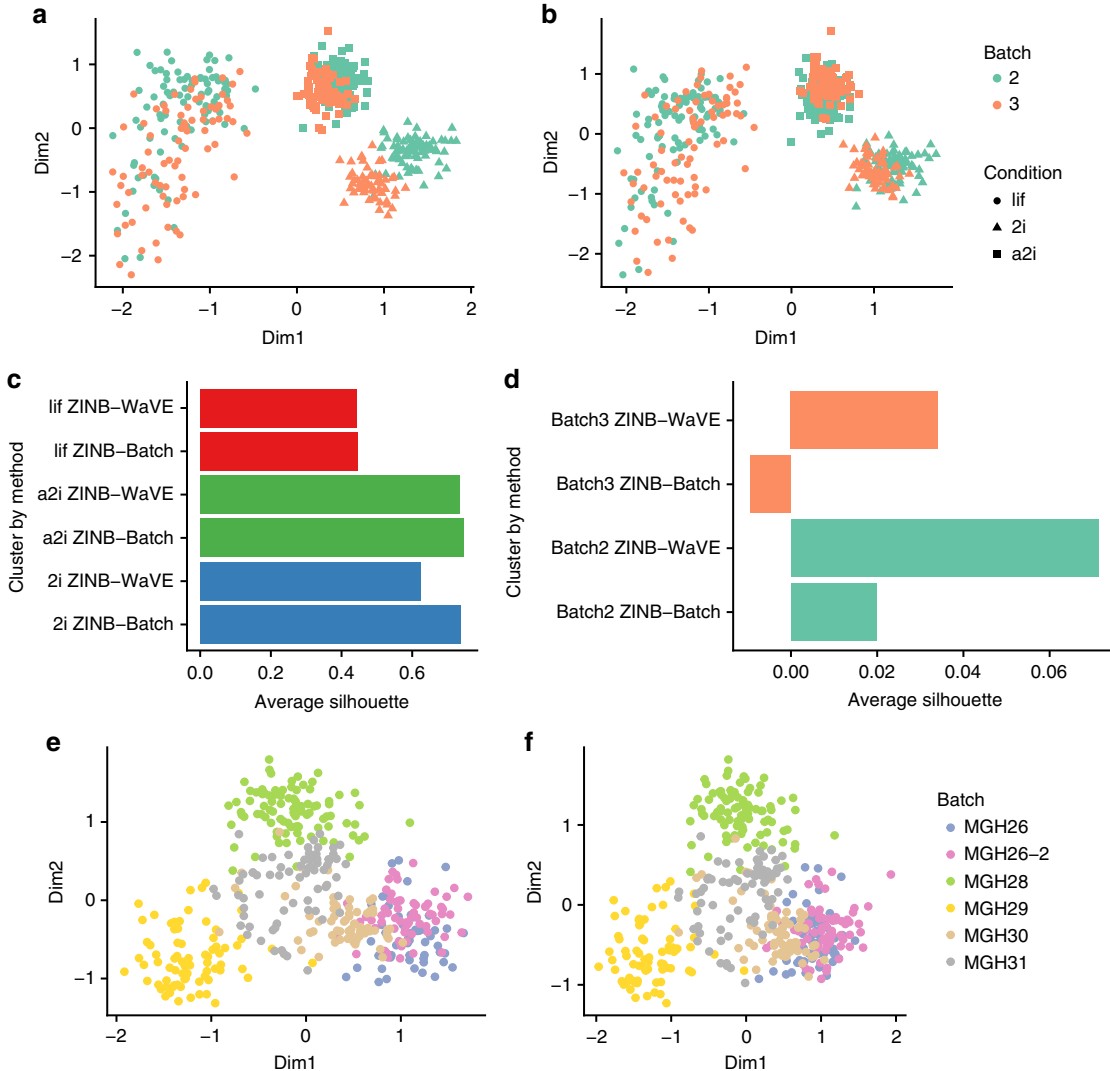

**Fig. 5** Accounting for batch effects in ZINB-WaVE. Upper panels provide two-dimensional representations of the mESC data, with cells color-coded by batch and shape reflecting culture conditions: **a** Default ZINB- WaVE with only sample-level intercept; **b** ZINB-WaVE with batch as known sample-level covariate. **c** Average silhouette widths by biological condition for ZINB-WaVE with and without batch covariate; **d** average silhouette widths by batch for ZINB-WaVE with and without batch covariate. Although the cells cluster primarily based on culture condition, batch effects are evident in **a**. Accounting for batch effects in the ZINB-WaVE model (**b**) leads to slightly better clustering by biological condition (**c**) and removes the clustering by batch (**d**). Note the difference in scale between the barplots of **c** and **d**. Lower panels provide two-dimensional representations of the glioblastoma data, with cells color-coded by batch: **e** Default ZINB-WaVE with only sample-level intercept; **f** ZINB-WaVE with total number of expressed genes as sample-level covariate. Despite the confounding between patient and batch, the addition of a covariate that captures the batch difference allows ZINB-WaVE to remove the batch effect without removing the patient effect

presence of confounding. ComBat was not able to correctly account for batch, removing the patient effects along with the batch effects (Supplementary Fig. 17a). Including the batch variable as a covariate in the ZINB-WaVE model led to similar, unsatisfactory results (Supplementary Fig. 17b). One key observation, recently made by Townes et al.[40], is that the detection rate is markedly different between the two batches of *MGH26* (Supplementary Fig. 17c): including the detection rate as a covariate in the ZINB-WaVE model led to the removal of the batch effect, while preserving the biological differences between patients (Fig. 5f). Note that this is analogous of the inclusion of the "cellular detection rate" in the MAST model[23]. Although this helped adjusting for batch effects, the confounding between patient and batch is still present in the data because of the poor experimental design and no normalization will be able to completely account for such confounding.

**Goodness-of-fit of ZINB-WaVE model**. We compared the goodness-of-fit of our ZINB-WaVE model to that of a negative binomial (NB) model (as implemented in edgeR[41]) and a hurdle model (as implemented in MAST[23]).

As previously noted in the literature, NB models, which are quite successful for bulk RNA-seq, are not appropriate for single-cell RNA-seq, as they cannot accommodate zero inflation. In particular, NB models poorly fit the data in terms of the overall mean count and zero probability (Supplementary Figs. 18 and 19) and appears to handle the excess of zeros by over-estimating the dispersion parameter (Supplementary Fig. 20).

We also examined the goodness-of-fit of the MAST hurdle model, which is specifically designed for scRNA-seq. However, a direct comparison of MAST with NB and ZINB is cumbersome, due to differences in parameterization. In particular, MAST models log2(TPM + 1) rather than counts and does not have a

dispersion parameter but only a variance parameter for the Gaussian component (see Methods for details). We found that MAST under-estimated the overall mean log2(TPM + 1) and over-estimated the zero probability, but had stable variance estimates over the observed proportion of zero counts (Supplementary Fig. 21).

By contrast, our ZINB model lead to better fit in terms of both the overall mean count and zero probability, as well as more stable estimators of the dispersion parameter (Supplementary Figs. 18–20).

**ZINB-WaVE estimators are asymptotically unbiased and robust.** We next turned to simulations to explore in greater detail the performance of ZINB-WaVE. First, we evaluated the ZINB-WaVE estimation procedure on simulated data from a ZINB distribution, to assess both accuracy under a correctly specified model and robustness to model misspecification. The approach involves computing maximum likelihood estimators (MLE) for the parameters of the model of Fig. 1, namely, $\alpha$, $\beta$, $\gamma$, and $W$, and hence $\mu$ and $\pi$. MLE are asymptotically unbiased and efficient estimators. However, because the likelihood function of our ZINB-WaVE model is not convex, our numerical optimization procedure may converge to a local maximum, rather than to the true MLE (see Methods). We observed that our estimators are asymptotically unbiased and have decreasing variance as the

number of cells $n$ increases (Supplementary Fig. 22). This suggests that our estimates are not far from the true MLE.

To assess the robustness of the ZINB-WaVE procedure, we examined bias and mean squared error (MSE) for estimators of $\mu$ and $\pi$, as well as the log-likelihood function, the Akaike information criterion (AIC), and the Bayesian information criterion (BIC), for models with misspecified number of unobserved covariates $K$ (i.e., number of columns of $W$), gene-level covariate matrix $V$, and dispersion parameter $\varphi$ (Fig. 6; Supplementary Figs. 23–25). In Fig. 6, the data were simulated with $K = 2$ unknown factors, $X = \mathbf{1}_n$, $V = \mathbf{1}_J$, and genewise dispersion. The model parameters were then estimated with $K = 1, 2, 3, 4$, both for a model that included a cell-level intercept ($V = \mathbf{1}_J$) and one that did not ($V = 0_J$). When the intercept was correctly included in the model, misspecification of $K$ (with $K > 2$) resulted in no or very small bias (Fig. 6a, b), small MSE (Fig. 6c, d), and a greater impact on AIC and especially BIC (Supplementary Fig. 23). When no intercept was included in the fitted model, the sensitivity to $K$ became more important. However, although the data were simulated with $K = 2$, specifying $K \geq 3$ led to small bias and low MSE (Fig. 6). This is likely because one column of $W$ acted as a de facto intercept, overcoming the absence of $V$ and explaining why AIC and BIC are minimized at $K = 4$ (Supplementary Fig. 23). In addition, we observed robustness of the results to the choice of dispersion parameter

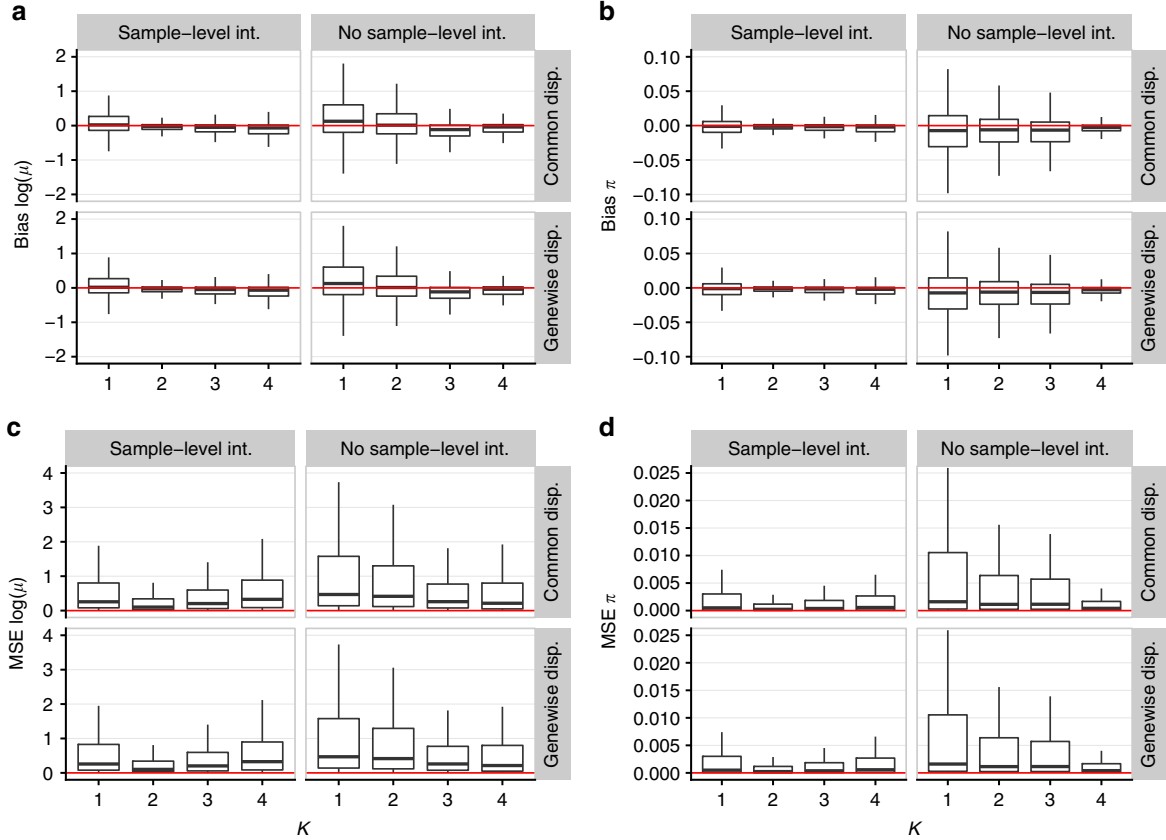

**Fig. 6** Bias and MSE for ZINB-WaVE estimation procedure. Panels show boxplots of **a** bias $\ln(\mu)$, **b** bias $\pi$, **c** MSE $\ln(\mu)$, and **d** MSE $\pi$ for ZINB-WaVE estimation procedure, as a function of the number of unknown covariates $K$. ZINB-WaVE was fit with $X = \mathbf{1}_n$, common/genewise dispersion, and with/without sample-level intercept (i.e., column of ones in gene-level covariate matrix $V$). For each gene and cell, bias and MSE were averaged over $B = 10$ data sets simulated from our ZINB-WaVE model, based on the S1/CA1 data set and with $n = 1000$ cells, $J = 1000$ genes, scaling of one for the ratio of within to between-cluster sums of squares ($b^2 = 1$), $K = 2$ unknown factors, zero fraction of about 80%, $X = \mathbf{1}_n$, cell-level intercept ($V = \mathbf{1}_J$), and genewise dispersion. The lower and upper hinges of the boxplots correspond to the first and third quartiles (the 25th and 75th percentiles), respectively. The upper/lower whisker extends from the hinge to the largest/smallest value no further than 1.5 × IQR from the hinge (where IQR is the inter-quartile range or difference between the third and first quartiles). Data beyond the whiskers are called outliers and are not plotted here. Supplementary Fig. 24 provides the same boxplots with individually plotted outliers

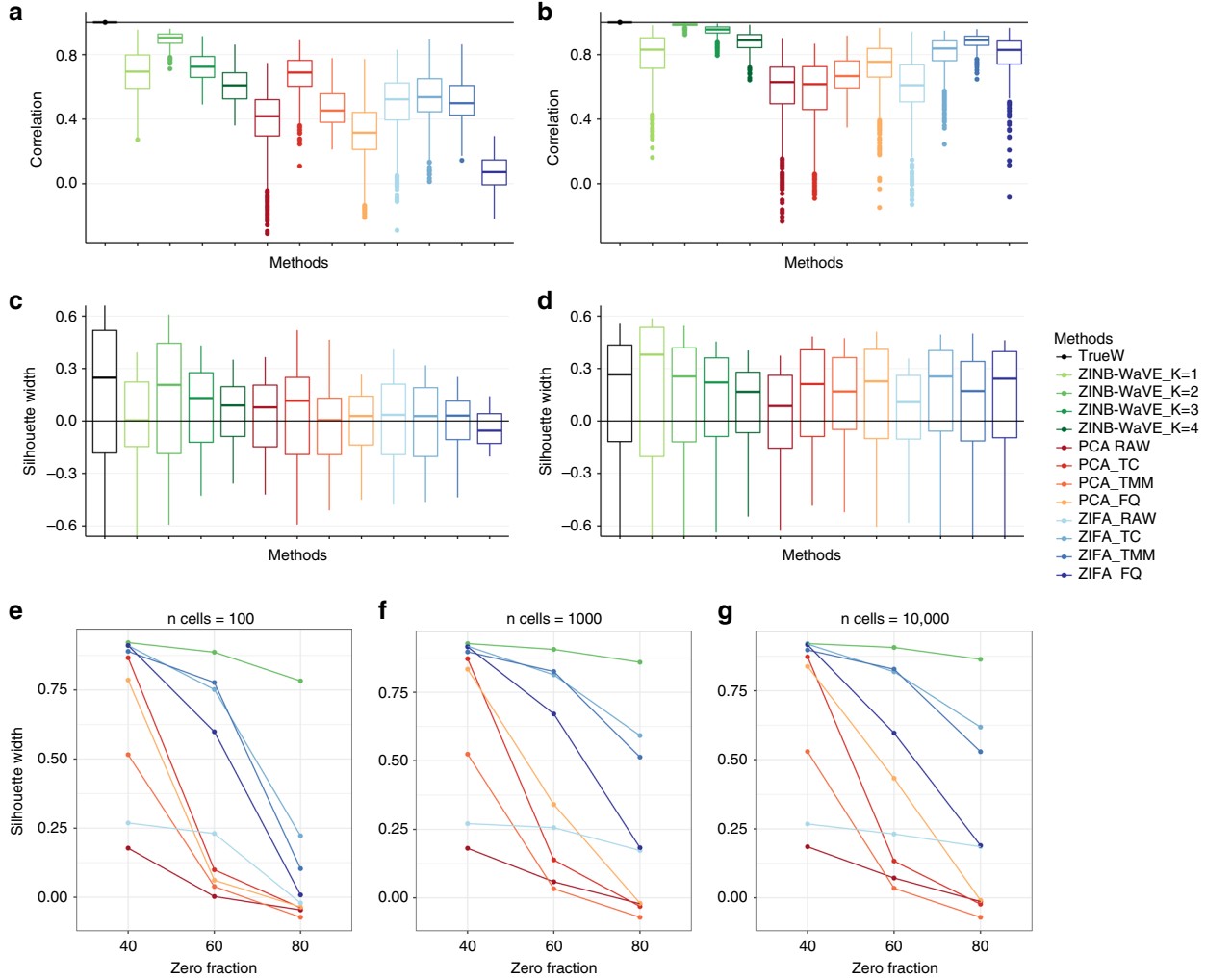

**Fig. 7** Between-sample distances and silhouette widths on simulated data. **a** Boxplots of correlations between between-sample distances based on true and estimated low-dimensional representations of the data for simulations based on the V1 data set. **b** Same as **a** for simulations based on the S1/CA1 data set. **c** Boxplots of silhouette widths for true clusters for simulations based on the V1 data set. **d** Same as **c** for simulations based on the S1/CA1 data set. For **a–d**, all data sets were simulated from our ZINB-WaVE model with $n = 1000$ cells, $J = 1000$ genes, "harder" clustering ($b^2 = 5$), $K = 2$ unknown factors, zero fraction of about 80%, $X = \mathbf{1}_n$, cell-level intercept ($V = \mathbf{1}_J$), and genewise dispersion. Each boxplot is based on $n$ values corresponding to each of the $n$ samples and defined as averages of correlations (**a**, **b**) or silhouette widths (**c**, **d**) over $B = 10$ simulations. See Supplementary Fig. 27 for the same scenario but with $n = 10,000$ cells and Supplementary Fig. 28 for additional scenarios. **e–g** Average silhouette widths (over $n$ samples and $B = 10$ simulations) for true clusters vs. zero fraction, for $n \in \{100; 1000; 10,000\}$ cells, for data sets simulated from the Lun & Marioni[42] model, with $C = 3$ clusters and equal number of cells per cluster. Although ZINB-WaVE was relatively robust to the sample size $n$ and zero fraction, the performance of PCA and ZIFA decreased with larger zero fraction. Between-sample distance matrices and silhouette widths were based on $W$ for ZINB-WaVE, the first two principal components for PCA, and the first two latent variables for ZIFA. ZINB-WaVE was applied with $X = \mathbf{1}_n$, $V = \mathbf{1}_J$, genewise dispersion, and $K \in \{1, 2, 3, 4\}$ (only $K = 2$ is shown in **e–g**. For PCA and ZIFA, different normalization methods were used. Colors correspond to the different methods

(genewise or common), even though the data were simulated with genewise dispersion (Fig. 6). The estimators for $\ln(\mu)$ and $\pi$ were unbiased over the whole range of mean expression and zero inflation probability (Supplementary Fig. 26).

**ZINB-WaVE is more accurate than state-of-the-art methods**. We next compared ZINB-WaVE to PCA and ZIFA, in terms of their ability to recover the true underlying low-dimensional signal and clustering structure. For data sets simulated from a ZINB model, our estimation procedure with correctly specified $K$ led to almost perfect correlation between distances in the true and estimated low-dimensional space (Fig. 7a, b). The correlation remained high even for misspecified $K$, in most cases higher than for PCA and ZIFA. ZINB-WaVE performed consistently well across a range of simulation scenarios, including different

numbers of cells $n$, different zero fractions, and varying cluster tightness (Supplementary Fig. 28). We observed a consistent ranking, although noisier, when the methods were compared in terms of silhouette widths (Fig. 7c, d).

As with the real data sets, the choice of normalization influenced the simulation results. Critically, there was not an overall best normalization method; rather, the performance of the normalization methods depended on the dimensionality reduction method and on intrinsic characteristics of the data, such as the fraction of zero counts and the number and tightness of the clusters (Fig. 7; Supplementary Fig. 28). For instance, TC normalization worked best for PCA on data simulated from the V1 data set (Fig. 7a, c), whereas FQ and TMM normalization worked best for PCA and ZIFA, respectively, on data simulated from the S1/CA1 data set (Fig. 7b, d).

It is important to note that the previous results were obtained for data simulated from the ZINB-WaVE model underlying our estimation procedure. It is hence not surprising that ZINB-WaVE outperformed its competitors. To provide a fairer comparison, we also assessed the methods on data simulated from the model proposed by Lun & Marioni[42]. Although this model is also based on a ZINB distribution, the distribution is parameterized differently and, in particular, does not involve regression on gene-level and sample-level covariates (see Methods).

When the data were simulated to have a moderate fraction of zeros (namely, 40%), all methods performed well in terms of average silhouette width (Fig. 7e–g) and precision and recall (Supplementary Fig. 29). However, the performance of PCA decreased dramatically with 60% of zeros, independently of the number of cells $n$. Although ZIFA worked well with 60% or fewer zeros, its performance too decreased at 80% of zeros, especially when only 100 cells were simulated (Fig. 7e). Conversely, the performance of ZINB-WaVE remained good even when simulating data with 80% of zeros, independently of the sample size.

## Discussion

Recent advances in single-cell technologies, such as droplet-based methods[2], make it easy and inexpensive to collect hundreds of thousands of scRNA-seq profiles, allowing researchers to study very complex biological systems at high resolution. The resulting libraries are often sequenced at extremely low depth (tens of thousands of reads per cell, only), making the corresponding read count data truly sparse. Hence, there is a growing need for developing reliable statistical methods that are scalable and that can account for zero inflation.

ZINB-WaVE is a general and flexible approach to extract low-dimensional signal from noisy, zero-inflated data, such as those from scRNA-seq experiments. We have shown with simulated and real data analyses that ZINB-WaVE leads to robust and unbiased estimators of the underlying biological signals. The better performance of ZINB-WaVE with respect to PCA comes at a computational cost, as we need to numerically optimize a non-convex likelihood function. However, we empirically found that the computing time was approximately linear in both the number of cells and the number of genes, and approximately quadratic in the number of latent factors (Supplementary Fig. 30). The algorithm benefits from parallelization on multicore machines and takes a few minutes on a modern laptop to converge for thousands of cells.

One major difference between ZINB-WaVE and previously proposed factor analysis models (such as PCA and ZIFA) is the ability to include sample-level and gene-level covariates. In particular, by including a column of ones in the gene-level covariate matrix, the corresponding cell-level intercept acts as a global-scaling normalization factor, allowing the modeling of raw count data, with no need for prior normalization.

However, there is no guarantee that the low-dimensional signal extracted by ZINB-WaVE is biologically relevant: If unwanted technical variation affects the data and is not accounted for in the model (or in prior normalization), the low-rank matrix $W$ inferred by ZINB-WaVE will capture such confounding effects. It is therefore important to explore the correlation between the latent factors estimated by our procedure and known measures of quality control that can be computed for scRNA-seq libraries, using, for instance, the Bioconductor R package scater[43] (see Fig. 2f). If one observes high correlation between one or more latent factors and some QC measures, it may be beneficial to include these QC measures as covariates in the model.

Several authors have recognized that high-dimensional genomic data are affected by a variety of unwanted technical effects (e.g., batch effects) that can be confounded with the biological signal of interest, and have proposed methods to account for such effects in either a supervised[38] or unsupervised way[27,44]. Recently, Lin et al.[45] proposed a model that can extend PCA to adjust for confounding factors. This model, however, does not seem to be ideal for zero-inflated count data. In the scRNA-seq literature, MAST[23] uses the inferred cellular detection rate to adjust for the main source of confounding, in a differential expression setting, but is not designed to infer low-dimensional signal.

The removal of batch effects is an important example of how including additional covariates in the ZINB-WaVE model may lead to better low-dimensional representations of the data. However, ZINB-WaVE is not limited to including batch effects, as other sample-level (e.g., QC metrics) and/or gene-level (e.g., GC-content) covariates may be included in the model. Although we did not find any compelling examples in which adding a gene-level covariate leads to improve signal extraction, it is interesting to note the relationship between GC-content and batch effects[46]. With large collaborative efforts, such as the Human Cell Atlas[47], on the horizon, we anticipate that the ability of our model to include gene-level covariates that can potentially help accounting for differences in protocols will prove important.

Although the low-dimensional signal inferred by ZINB-WaVE can be used to visually inspect hidden structure in the data, visualization is not the main point of our proposed method. The low-dimensional factors are intended to be the closest possible approximation to the true signal, which is assumed to be intrinsically low-dimensional. Such a low-dimensional representation can be used in downstream analyses, such as clustering or pseudotime ordering of the cells[15].

Visualization of high-dimensional data sets is an equally important area of research and many algorithms are available, among which t-SNE[25] has become the most popular for scRNA-seq data. Recently, Wang et al.[48] have proposed a novel visualization algorithm that can account for zero inflation and showed improvement over t-SNE. As t-SNE takes as input a matrix of cell pairwise distances, which may be noisy in high dimensions, a typical pipeline involves computing such distances in PCA space, selecting, for example, the first 50 PCs. An alternative approach is to derive such distances from the low-dimensional space defined by the factors inferred by ZINB-WaVE. This strategy was used effectively in Fig. 3c to visualize the PBMC data set.

In this article, we have focused on an unsupervised setting, where the goal is to extract a low-dimensional signal from noisy zero-inflated data. However, our proposed ZINB model is more general and can be used, in principle, for supervised differential expression analysis, where the parameters of interest are regression coefficients $\beta$ corresponding to known sample-level covariates in the matrix $X$ (e.g., cell type, treatment/control status). Differentially expressed genes may be identified via likelihood ratio tests or Wald tests, with standard errors of estimators of $\beta$ obtained from the Hessian matrix of the likelihood function. In addition, posterior dropout probabilities can be readily derived from the model and used as weights to unlock standard bulk RNA-seq methods[49], such as edgeR[41]. We envision a future version of the zinbwave package with this added capability.

## Methods

**ZINB-WaVE model**. For any $\mu \geq 0$ and $\theta > 0$, let $f_{NB}(\cdot; \mu, \theta)$ denote the probability mass function (PMF) of the negative binomial (NB) distribution with mean $\mu$ and inverse dispersion parameter $\theta$, namely:

$$f_{\mathrm{NB}}(y; \mu, \theta) = \frac{\Gamma(y + \theta)}{\Gamma(y + 1)\Gamma(\theta)} \left( \frac{\theta}{\theta + \mu} \right)^{\theta} \left( \frac{\mu}{\mu + \theta} \right)^{y}, \quad \forall y \in \mathbb{N}. \tag{1}$$

Note that another parametrization of the NB PMF is in terms of the dispersion parameter $\varphi = \theta^{-1}$ (although $\theta$ is also sometimes called dispersion parameter in

the literature). In both cases, the mean of the NB distribution is $\mu$ and its variance is:

$$\sigma^2 = \mu + \frac{\mu^2}{\theta} = \mu + \phi\mu^2. \qquad (2)$$

In particular, the NB distribution boils down to a Poisson distribution when $\varphi = 0 \Leftrightarrow \theta = +\infty$.

For any $\pi \in [0, 1]$, let $f_{\mathrm{ZINB}}(\cdot\, ; \mu, \theta, \pi)$ be the PMF of the ZINB distribution given by:

$$f_{\mathrm{ZINB}}(y; \mu, \theta, \pi) = \pi\delta_0(y) + (1-\pi)f_{\mathrm{NB}}(y; \mu, \theta), \quad \forall y \in \mathbb{N}, \qquad (3)$$

where $\delta_0(\cdot)$ is the Dirac function. Here, $\pi$ can be interpreted as the probability that a 0 is observed instead of the actual count, resulting in an inflation of zeros compared to the NB distribution, hence the name ZINB.

Given $n$ samples (typically, $n$ single cells) and $J$ features (typically, $J$ genes) that can be counted for each sample, let $Y_{ij}$ denote the count of feature $j$ (for $j = 1,\ldots, J$) for sample $i$ ($i = 1,\ldots, n$). To account for various technical and biological effects frequent, in particular, in single-cell sequencing technologies, we model $Y_{ij}$ as a random variable following a ZINB distribution with parameters $\mu_{ij}$, $\theta_{ij}$, and $\pi_{ij}$, and consider the following regression models for the parameters:

$$\begin{aligned} \ln\left(\mu_{ij}\right) &= \left(X\beta_\mu + (V\gamma_\mu)^\top + W\alpha_\mu + O_\mu\right)_{ij}, \\ \mathrm{logit}\left(\pi_{ij}\right) &= \left(X\beta_\pi + (V\gamma_\pi)^\top + W\alpha_\pi + O_\pi\right)_{ij}, \\ \ln\left(\theta_{ij}\right) &= \zeta_j, \end{aligned} \qquad (4)$$

where

$$\mathrm{logit}(\pi) = \ln\left(\frac{\pi}{1-\pi}\right)$$

and elements of the regression models are as follows.

$X$ is a known $n \times M$ matrix corresponding to $M$ cell-level covariates and $\beta = (\beta_\mu, \beta_\pi)$ its associated $M \times J$ matrices of regression parameters. $X$ can typically include covariates that induce variation of interest, such as cell types, or covariates that induce unwanted variation, such as batch or quality control measures. It can also include a constant column of ones, $\mathbf{1}_n$, to account for gene-specific intercepts.

$V$ is a known $J \times L$ matrix corresponding to $J$ gene-level covariates, such as gene length or GC-content, and $\gamma = (\gamma_\mu, \gamma_\pi)$ its associated $L \times n$ matrices of regression parameters. $V$ can also include a constant column of ones, $\mathbf{1}_J$, to account for cell-specific intercepts, such as size factors representing differences in library sizes.

$W$ is an unobserved $n \times K$ matrix corresponding to $K$ unknown cell-level covariates, which could be of "unwanted variation" as in RUV[27,28] or of interest (such as cell type), and $\alpha = (\alpha_\mu, \alpha_\pi)$ its associated $K \times J$ matrices of regression parameters.

$O_\mu$ and $O_\pi$ are known $n \times J$ matrices of offsets. $\zeta \in \mathbb{R}^J$ is a vector of gene-specific dispersion parameters on the log scale. This model deserves a few comments.

By default, $X$ and $V$ contain a constant column of ones, to account, respectively, for gene-specific (e.g., baseline expression level) and cell-specific (e.g., library size) variation. In that case, $X$ and $V$ are of the form $X = [\mathbf{1}_n, X^0]$ and $V = [\mathbf{1}_J, V^0]$ and we can similarly decompose the corresponding parameters as $\beta = [\beta^1, \beta^0]$ and $\gamma = [\gamma^1, \gamma^0]$, where $\beta^1 \in \mathbb{R}^{1 \times J}$ is a vector of gene-specific intercepts and $\gamma^1 \in \mathbb{R}^{1 \times n}$ a vector of cell-specific intercepts. The representation $\mathbf{1}_n\beta^1 + (\mathbf{1}_J\gamma^1)^\top$ is then not unique, but could be made unique by adding a constant and constraining $\beta^1$ and $\gamma^1$ to each have elements summing to zero.

Although $W$ is the same, the matrices $X$ and $V$ could differ in the modeling of $\mu$ and $\pi$, if we assume that some known factors do not affect both $\mu$ and $\pi$. To keep notation simple and consistent, we use the same matrices, but will implicitly assume that some parameters may be constrained to be 0 if needed.

By allowing the models to differ for $\mu$ and $\pi$, we can model and test for differential expression in terms of either the NB mean or the ZI probability.

We limit ourselves to a gene-dependent dispersion parameter. More complicated models for $\theta_{ij}$ could be investigated, such as a model similar to $\mu_{ij}$ or a functional of the form $\theta_{ij} = f(\mu_{ij})$, but we restrict ourselves to a simpler model that has been shown to be largely sufficient in bulk RNA-seq analysis.

**ZINB-WaVE estimation procedure**. The input to the model are the matrices $X$, $V$, $O_\mu$, and $O_\pi$ and the integer $K$; the parameters to be inferred are $\beta = (\beta_\mu, \beta_\pi)$, $\gamma = (\gamma_\mu, \gamma_\pi)$, $W$, $\alpha = (\alpha_\mu, \alpha_\pi)$, and $\zeta$. Given an $n \times J$ matrix of counts $Y$, the log-likelihood function is

$$\ell(\beta, \gamma, W, \alpha, \zeta) = \sum_{i=1}^{n}\sum_{j=1}^{J} \ln f_{\mathrm{ZINB}}\left(Y_{ij}; \mu_{ij}, \theta_{ij}, \pi_{ij}\right), \qquad (5)$$

where $\mu_{ij}$, $\theta_{ij}$, and $\pi_{ij}$ depend on $(\beta, \gamma, W, \alpha, \zeta)$ through Eq. (4).

To infer the parameters, we follow a penalized maximum likelihood approach, by trying to solve

$$\max_{\beta,\gamma,W,\alpha,\zeta}\{\ell(\beta, \gamma, W, \alpha, \zeta) - \mathrm{Pen}(\beta, \gamma, W, \alpha, \zeta)\},$$

where $\mathrm{Pen}(\cdot)$ is a regularization term to reduce overfitting and improve the numerical stability of the optimization problem in the setting of many parameters. For nonnegative regularization parameters $(\epsilon_\beta, \epsilon_\gamma, \epsilon_W, \epsilon_\alpha, \epsilon_\zeta)$, we set

$$\mathrm{Pen}(\beta, \gamma, W, \alpha, \zeta) = \frac{\epsilon_\beta}{2}\|\beta^0\|^2 + \frac{\epsilon_\gamma}{2}\|\gamma^0\|^2 + \frac{\epsilon_W}{2}\|W\|^2 + \frac{\epsilon_\alpha}{2}\|\alpha\|^2 + \frac{\epsilon_\zeta}{2}\mathrm{Var}(\zeta),$$

where $\beta^0$ and $\gamma^0$ denote the matrices $\beta$ and $\gamma$ without the rows corresponding to the intercepts if an unpenalized intercept is included in the model, $\|\cdot\|$ is the Frobenius matrix norm ($\|A\| = \sqrt{\mathrm{tr}(A^*A)}$, where $A^*$ denotes the conjugate transpose of $A$), and $\mathrm{Var}(\zeta) = 1/(J-1)\sum_{i=1}^{J}\left(\zeta_i - \left(\sum_{j=1}^{J}\zeta_j\right)/J\right)^2$ is the variance of the elements of $\zeta$ (using the unbiased sample variance statistic). The penalty tends to shrink the estimated parameters to 0, except for the cell and gene-specific intercepts which are not penalized and the dispersion parameters which are not shrunk towards 0 but instead towards a constant value across genes. Note also that the likelihood only depends on $W$ and $\alpha$ through their product $R = W\alpha$ and that the penalty ensures that at the optimum $W$ and $\alpha$ have the structure described in the following result, which generalizes standard results such as Srebro et al.[50] (Lemma 1) and Mazumder et al.[51] (Lemma 6).

*Lemma 1:* For any matrix $R$ and positive scalars $s$ and $t$, the following holds:

$$\min_{S,T:R=ST}\frac{1}{2}\left(s\|S\|^2 + t\|T\|^2\right) = \sqrt{st}\|R\|_*,$$

where $\|A\|_* = \mathrm{tr}\left(\sqrt{A^*A}\right)$. If $R = R_L R_\Sigma R_R$ is a singular value decomposition (SVD) of $R$, then a solution to this optimization problem is:

$$S = \left(\frac{t}{s}\right)^{\frac{1}{4}}R_L R_\Sigma^{\frac{1}{2}}, \; T = \left(\frac{s}{t}\right)^{\frac{1}{4}}R_\Sigma^{\frac{1}{2}}R_R.$$

Proof: Let $\tilde{S} = \sqrt{s}S$, $\tilde{T} = \sqrt{t}T$, and $\tilde{R} = \sqrt{st}R$. Then, $\|\tilde{S}\|^2 = s\|S\|^2$, $\|\tilde{T}\|^2 = t\|T\|^2$, and $\tilde{S}\tilde{T} = \sqrt{st}ST$, so that the optimization problem is equivalent to:

$$\min_{\tilde{S},\tilde{T}:\tilde{S}\tilde{T}=\tilde{R}}\frac{1}{2}\left(\|\tilde{S}\|^2 + \|\tilde{T}\|^2\right),$$

which by Mazumder et al.[51] (Lemma 6) has optimum value $\|\tilde{R}\|_* = \sqrt{st}\|R\|_*$ reached at $\tilde{S} = \tilde{R}_L\tilde{R}_\Sigma^{\frac{1}{2}}$ and $\tilde{T} = \tilde{R}_\Sigma^{\frac{1}{2}}\tilde{R}_R$, where $\tilde{R}_L\tilde{R}_\Sigma\tilde{R}_R$ is a SVD of $\tilde{R}$. Observing that $\tilde{R}_L = R_L$, $\tilde{R}_R = R_R$, and $\tilde{R}_\Sigma = \sqrt{st}R_\Sigma$, gives that a solution of the optimization problem is $S = s^{-1/2}\tilde{S} = s^{-1/2}R_L(st)^{1/4}R_\Sigma^{1/2} = (t/s)^{1/4}R_L R_\Sigma^{1/2}$. A similar argument for $T$ concludes the proof.

This lemma implies in particular that at any local maximum of the penalized log-likelihood, $W$ and $\alpha^\top$ have orthogonal columns, which is useful for visualization or interpretation of latent factors.

To balance the penalties applied to the different matrices in spite of their different sizes, a natural choice is to fix $\epsilon > 0$ and set

$$\epsilon_\beta = \frac{\epsilon}{J}, \; \epsilon_\gamma = \frac{\epsilon}{n}, \; \epsilon_W = \frac{\epsilon}{n}, \; \epsilon_\alpha = \frac{\epsilon}{J}, \; \epsilon_\zeta = \epsilon.$$

In particular, from Lemma 1, we easily deduce the following characterization of the penalty on $W$ and $\alpha$, which shows that the entries in the matrices $W$ and $\alpha$ have similar standard deviation after optimization:

Corollary 1: For any $n \times J$ matrix $R$ and positive scalars $\epsilon$, the following holds.

$$\min_{W,\alpha:R=W\alpha}\frac{\epsilon}{2}\left(\frac{1}{n}\|W\|^2 + \frac{1}{J}\|\alpha\|^2\right) = \frac{\epsilon}{\sqrt{nJ}}\|R\|_*.$$

If $R = R_L R_\Sigma R_R$ is a SVD decomposition of $R$, then a solution to this optimization problem is:

$$W = \left(\frac{n}{J}\right)^{\frac{1}{4}}R_L R_\Sigma^{\frac{1}{2}}, \; T = \left(\frac{J}{n}\right)^{\frac{1}{4}}R_\Sigma^{\frac{1}{2}}R_R.$$

In particular, for any $i = 1, \ldots, \min(n, J)$,

$$\frac{1}{n}\sum_{j=1}^{n}W_{j,i}^2 = \frac{1}{J}\sum_{j=1}^{J}\alpha_{i,j}^2 = \frac{[R_\Sigma]_{i,i}}{\sqrt{nJ}}.$$

The penalized likelihood is however not concave, making its maximization computationally challenging. We instead find a local maximum, starting from a smart initialization and iterating a numerical optimization scheme until local convergence, as described below.

**Initialization:** To initialize the set of parameters we approximate the count distribution by a log-normal distribution and explicitly separate zero and non-zero values, as follows:

1. Set $\mathcal{P} = \{(i,j) : Y_{ij} > 0\}$.
2. Set $L_{ij} = \ln(Y_{ij}) - (O_\mu)_{ij}$ for all $(i,j) \in \mathcal{P}$.
3. Set $\hat{Z}_{ij} = 0$ if $(i,j) \in \mathcal{P}$, $\hat{Z}_{ij} = 1$ otherwise.
4. Estimate $\beta_\mu$ and $\gamma_\mu$ by solving the convex ridge regression problem:

$$\min_{\beta_\mu, \gamma_\mu} \sum_{(i,j)\in\mathcal{P}} \left( L_{ij} - (X\beta_\mu)_{ij} - (V\gamma_\mu)_{ji} \right)^2 + \frac{\epsilon_\beta}{2} \left\| \beta_\mu^0 \right\|^2 + \frac{\epsilon_\gamma}{2} \left\| \gamma_\mu^0 \right\|^2.$$

This is a standard ridge regression problem, but with a potentially huge design matrix, with up to $nJ$ rows and $MJ + nL$ columns. To solve it efficiently, we alternate the estimation of $\beta_\mu$ and $\gamma_\mu$. Specifically, we initialize parameter values as:

$$\hat{\beta}_\mu \leftarrow 0, \quad \hat{\gamma}_\mu \leftarrow 0$$

and repeat the following two steps a few times (or until convergence):

a. Optimization in $\gamma_\mu$, which can be performed independently and in parallel for each cell:

$$\hat{\gamma}_\mu \in \underset{\gamma_\mu}{\operatorname{argmin}} \sum_{(i,j)\in\mathcal{P}} \left( L_{ij} - (X\hat{\beta}_\mu)_{ij} - (V\gamma_\mu)_{ji} \right)^2 + \frac{\epsilon_\gamma}{2} \left\| \gamma_\mu^0 \right\|^2.$$

b. Optimization in $\beta_\mu$, which can be performed independently and in parallel for each gene:

$$\hat{\beta}_\mu \in \underset{\beta_\mu}{\operatorname{argmin}} \sum_{(i,j)\in\mathcal{P}} \left( L_{ij} - (V\hat{\gamma}_\mu)_{ji} - (X\beta_\mu)_{ij} \right)^2 + \frac{\epsilon_\beta}{2} \left\| \beta_\mu^0 \right\|^2.$$

5. Estimate $W$ and $\alpha_\mu$ by solving

$$(\hat{W}, \hat{\alpha}_\mu) \in \underset{W, \alpha_\mu}{\operatorname{argmin}} \sum_{(i,j)\in\mathcal{P}} \left( L_{ij} - (X\hat{\beta}_\mu)_{ij} - (V\hat{\gamma}_\mu)_{ji} - (W\alpha_\mu)_{ij} \right)^2 + \frac{\epsilon_W}{2} \|W\|^2 + \frac{\epsilon_\alpha}{2} \|\alpha_\mu\|^2.$$

Denoting by $D = L - X\hat{\beta} - (V\hat{\gamma})^\top$, this problem can be rewritten as:

$$\min_{W, \alpha} \|D - W\alpha\|_\mathcal{P}^2 + \frac{1}{2} \left( \epsilon_W \|W\|^2 + \epsilon_\alpha \|\alpha\|^2 \right),$$

where $\|A\|_\mathcal{P}^2 = \sum_{(i,j)\in\mathcal{P}} A_{ij}^2$. By Lemma 1, if $K$ is large enough, one can first solve the convex optimization problem:

$$\hat{R} \in \underset{R:\operatorname{rank}(R)\leq K}{\operatorname{argmin}} \|D - R\|_\mathcal{P}^2 + \sqrt{\epsilon_W \epsilon_\alpha} \|R\|_* \qquad (6)$$

and set

$$W = \left( \frac{\epsilon_\alpha}{\epsilon_W} \right)^{\frac{1}{4}} R_L R_\Sigma^{\frac{1}{2}}, \quad \alpha = \left( \frac{\epsilon_W}{\epsilon_\alpha} \right)^{\frac{1}{4}} R_\Sigma^{\frac{1}{2}} R_R,$$

where $\hat{R} = R_L R_\Sigma R_R$ is the SVD of $\hat{R}$. This solution is exact when $K$ is at least equal to the rank of the solution of the unconstrained problem (6), which we solve with the `softImpute::softImpute()` function[51]. If $K$ is smaller, then (6) becomes a non-convex optimization problem whose global optimum may be challenging to find. In that case we also use the rank-constrained version of `softImpute::softImpute()` to obtain a good local optimum.

6. Estimate $\beta_\pi$, $\gamma_\pi$, and $\alpha_\pi$ by solving the regularized logistic regression problem:

$$\min_{(\beta_\pi, \gamma_\pi, \alpha_\pi)} \sum_{(i,j)} \left[ -\hat{Z}_{ij} \left( X\beta_\pi + (V\gamma_\pi)^\top + \hat{W}\alpha_\pi \right)_{ij} \right.$$
$$\left. + \ln\left( 1 + e^{\left( X\beta_\pi + (V\gamma_\pi)^\top + \hat{W}\alpha_\pi \right)_{ij}} \right) \right] + \frac{\epsilon_\beta}{2} \|\beta_\pi\|^2 + \frac{\epsilon_\gamma}{2} \|\gamma_\pi\|^2 + \frac{\epsilon_\alpha}{2} \|\alpha_\pi\|^2. \qquad (7)$$

This is a standard ridge logistic regression problem, but with a potentially huge design matrix, with up to $nJ$ rows and $MJ + nL$ columns. To solve it efficiently, we alternate the estimation of $\beta_\pi$, $\gamma_\pi$, and $\alpha_\pi$. Specifically, we initialize parameter values as:

$$\hat{\beta}_\pi \leftarrow 0, \quad \hat{\gamma}_\pi \leftarrow 0, \quad \hat{\alpha}_\pi \leftarrow 0$$

and repeat the following two steps a few times (or until convergence):

a. Optimization in $\gamma_\pi$:

$$\hat{\gamma}_\pi \in \underset{\gamma_\pi}{\operatorname{argmin}} \sum_{(i,j)} \left[ -\hat{Z}_{ij} \left( X\hat{\beta}_\pi + (V\gamma_\pi)^\top + \hat{W}\hat{\alpha}_\pi \right)_{ij} \right.$$
$$\left. + \ln\left( 1 + e^{\left( X\hat{\beta}_\pi + (V\gamma_\pi)^\top + \hat{W}\hat{\alpha}_\pi \right)_{ij}} \right) \right] + \frac{\epsilon_\gamma}{2} \|\gamma_\pi\|^2. \qquad (8)$$

Note that this problem can be solved for each cell ($i$) independently and in parallel. When there is no gene covariate besides the constant intercept, the problem is easily solved by setting $(\hat{\gamma}_\pi)_i$ to the logit of the proportion of zeros in each cell.

b. Optimization in $\beta_\pi$ and $\alpha_\pi$:

$$(\hat{\beta}_\pi, \hat{\alpha}_\pi) \in \underset{(\beta_\pi, \alpha_\pi)}{\operatorname{argmin}} \sum_{(i,j)} \left[ -\hat{Z}_{ij} \left( X\beta_\pi + (V\hat{\gamma}_\pi)^\top + \hat{W}\alpha_\pi \right)_{ij} \right.$$
$$\left. + \ln\left( 1 + e^{\left( X\beta_\pi + (V\hat{\gamma}_\pi)^\top + \hat{W}\alpha_\pi \right)_{ij}} \right) \right] + \frac{\epsilon_\beta}{2} \|\beta_\pi\|^2 + \frac{\epsilon_\alpha}{2} \|\alpha_\pi\|^2. \qquad (9)$$

7. Initialize $\hat{\zeta} = 0$.

**Optimization:** After initialization, we maximize locally the penalized log-likelihood by alternating optimization over the dispersion parameters and left and right-factors, iterating the following steps until convergence:

1. Dispersion optimization:

$$\hat{\zeta} \leftarrow \underset{\zeta}{\operatorname{argmax}} \left\{ \ell(\hat{\beta}, \hat{\gamma}, \hat{W}, \hat{\alpha}, \zeta) - \frac{\epsilon_\zeta}{2} \operatorname{Var}(\zeta) \right\}.$$

To solve this problem, we start by estimating a common dispersion parameter for all the genes, by maximizing the objective function under the constraint that $\operatorname{Var}(\zeta) = 0$; in practice, we use a derivative-free one-dimensional optimization vector over the range $[-50, 50]$. We then optimize the objective function by a quasi-Newton optimization scheme starting from the constant solution found by the first step. To derive the gradient of the objective function used by the optimization procedure, note that the derivative of the NB log-density is:

$$\frac{\partial}{\partial\theta} \ln f_{\text{NB}}(y; \mu, \theta) = \Psi(y + \theta) - \Psi(\theta) + \ln\theta + 1 - \ln(\mu + \theta) - \frac{y + \theta}{\mu + \theta},$$

where $\Psi(z) = \Gamma'(z)/\Gamma(z)$ is the digamma function. We therefore get the derivative of the ZINB density as follows, for any $\pi \in [0, 1]$:

-If $y > 0$, $f_{\text{ZINB}}(y; \mu, \theta, \pi) = (1 - \pi)f_{\text{NB}}(y; \mu, \theta)$ therefore

$$\frac{\partial}{\partial\theta} \ln f_{\text{ZINB}}(y; \mu, \theta) = \Psi(y + \theta) - \Psi(\theta) + \ln\theta + 1 - \ln(\mu + \theta) - \frac{y + \theta}{\mu + \theta}.$$

-For $y = 0$, $\frac{\partial}{\partial\theta} \ln f_{\text{NB}}(0; \mu, \theta) = \ln\theta + 1 - \ln(\mu + \theta) - \frac{\theta}{\mu + \theta}$, therefore

$$\frac{\partial}{\partial\theta} \ln f_{\text{ZINB}}(y; \mu, \theta) = \frac{\ln\theta + 1 - \ln(\mu + \theta) - \frac{\theta}{\mu + \theta}}{1 + \frac{\pi(\mu + \theta)^\theta}{(1 - \pi)\theta^\theta}}.$$

The derivative of the objective function w.r.t. $\zeta_j$, for $j = 1, \ldots, J$, is then easily obtained by

$$\sum_{i=1}^{n} \theta_j \frac{\partial}{\partial\theta} \ln f_{\text{ZINB}}\left( y_{ij}; \mu_{ij}, \theta_j \right) - \frac{\epsilon_\zeta}{J - 1} \left( \zeta_j - \frac{1}{J} \sum_{k=1}^{J} \zeta_k \right).$$

(Note that the $J - 1$ term in the denominator comes from the use of the unbiased sample variance statistic in the penalty for $\zeta$.)

2. Left-factor (cell-specific) optimization:

$$(\hat{\gamma}, \hat{W}) \leftarrow \underset{(\gamma, W)}{\operatorname{argmax}} \left\{ \ell(\hat{\beta}, \gamma, W, \hat{\alpha}, \hat{\zeta}) - \frac{\epsilon_\gamma}{2} \|\gamma^0\|^2 - \frac{\epsilon_W}{2} \|W\|^2 \right\}. \qquad (10)$$

Note that this optimization can be performed independently and in parallel for each cell $i = 1, \ldots, n$. For this purpose, we consider a subroutine `solveZinbRegression` ($y$, $A_\mu$, $B_\mu$, $C_\mu$, $A_\pi$, $B_\pi$, $C_\pi$, $C_\theta$) to find a set of vectors $(a_\mu, a_\pi, b)$ that locally maximize the log-likelihood of a ZINB model for a vector of counts $y$ parametrized as follows:

$$\ln(\mu) = A_\mu a_\mu + B_\mu b + C_\mu,$$
$$\operatorname{logit}(\pi) = A_\pi a_\pi + B_\pi b + C_\pi,$$
$$\ln(\theta) = C_\theta.$$

We give more details on how to solve `solveZinbRegression` in the next section. To solve (10) for cell $i$ we call `solveZinbRegression` with the

following parameters:

$$
\begin{cases}
a_\mu = \gamma_\mu[.,i] \\
a_\pi = \gamma_\pi[.,i] \\
b = W[i,]^\top \\
y = Y[i,]^\top \\
A_\mu = V_\mu \\
B_\mu = \alpha_\mu^\top \\
C_\mu = \left(X_\mu[i,.]\beta_\mu + O_\mu[i,.]\right)^\top \\
A_\pi = V_\pi \\
B_\pi = \alpha_\pi^\top \\
C_\pi = \left(X_\pi[i,.]\beta_\pi + O_\pi[i,.]\right)^\top \\
C_\theta = \zeta
\end{cases}.
$$

3. Right-factor (gene-specific) optimization:

$$
(\hat{\beta},\hat{\alpha}) \leftarrow \underset{(\beta,\alpha)}{\mathrm{argmax}}\left\{\ell(\beta,\hat{\gamma},\hat{W},\alpha,\hat{\zeta}) - \frac{\epsilon_\beta}{2}\|\beta^0\|^2 - \frac{\epsilon_\alpha}{2}\|\alpha\|^2\right\}.
$$

Note that this optimization can be performed independently and in parallel for each gene $j=1,\dots,J$, by calling

`solveZinbRegression` with the following parameters:

$$
\begin{cases}
a_\mu &= (\beta_\mu[.,j];\alpha_\mu[.,j]) \\
a_\pi &= (\beta_\pi[.,j];\alpha_\pi[.,j]) \\
b &= \emptyset \\
y &= Y[.,j] \\
A_\mu &= [X_\mu,W] \\
B_\mu &= \emptyset \\
C_\mu &= (V_\mu[j,.]\gamma_\mu)^\top + O_\mu[.,j] \\
A_\pi &= [X_\pi,W] \\
B_\pi &= \emptyset \\
C_\pi &= (V_\pi[j,.]\gamma_\pi)^\top + O_\pi[.,j] \\
C_\theta &= \zeta_j \mathbf{1}_n
\end{cases}.
$$

4. Orthogonalization:

$$
(\hat{W},\hat{\alpha}) \leftarrow \underset{(W,\alpha):W\alpha=\hat{W}\hat{\alpha}}{\mathrm{argmin}} \frac{1}{2}\left(\epsilon_W\|W\|^2 + \epsilon_\alpha\|\alpha\|^2\right).
$$

This is obtained by applying Lemma 1, starting from an SVD decomposition of the current $\hat{W}\hat{\alpha}$. Note that this step not only allows to maximize locally the penalized log-likelihood, but also ensures that the columns of $W$ stay orthogonal to each other during optimization.

**Solving.** `solveZinbRegression` Given a $N$-dimensional matrix of counts $y \in \mathbb{N}^N$, and matrix $A_\mu \in \mathbb{R}^{N\times p}$, $B_\mu \in \mathbb{R}^{N\times r}$, $C_\mu \in \mathbb{R}^N$, $A_\pi \in \mathbb{R}^{N\times q}$, $B_\pi \in \mathbb{R}^{N\times r}$, $C_\pi \in \mathbb{R}^N$, and $C_\theta \in \mathbb{R}^N$, for some integers $p,q,r$, the function attempts to find parameters $(a_\mu, a_\pi, b) \in \mathbb{R}^p \times \mathbb{R}^q \times \mathbb{R}^r$ that maximize the ZINB log-likelihood of $y$ with parameters:

$$
\begin{aligned}
\ln(\mu) &= A_\mu a_\mu + B_\mu b + C_\mu, \\
\mathrm{logit}(\pi) &= A_\pi a_\pi + B_\pi b + C_\pi, \\
\ln(\theta) &= C_\theta.
\end{aligned}
$$

Starting from an initial guess (as explained in the different steps above), we perform a local minimization of this function $F(a_\mu, a_\pi, b)$ using the Broyden–Fletcher–Goldfarb–Shanno (BFGS) quasi-Newton method. Let us now give more details on how the gradient of $F$ is computed.

Given a single count $y$ (i.e., $N=1$), we first explicit the derivatives of the log-likelihood of $y$ with respect to the $(\mu,\pi)$ parameters of the ZINB distribution. We first observe that

$$
\frac{\partial}{\partial\mu}\ln f_{NB}(y;\mu,\theta) = \frac{y}{\mu} - \frac{y+\theta}{\mu+\theta},
$$

and that by definition of the ZINB distribution the following holds:

$$
\begin{aligned}
\frac{\partial}{\partial\mu}\ln f_{ZINB}(y;\mu,\theta,\pi) &= \frac{(1-\pi)f_{NB}(y;\mu,\theta)\frac{\partial}{\partial\mu}\ln f_{NB}(y;\mu,\theta)}{f_{ZINB}(y;\mu,\theta,\pi)}, \\
\frac{\partial}{\partial\pi}\ln f_{ZINB}(y;\mu,\theta,\pi) &= \frac{\delta_0(y)-f_{NB}(y;\mu,\theta)}{f_{ZINB}(y;\mu,\theta,\pi)}.
\end{aligned}
$$

Let us explicit these expressions, depending on whether or not $y$ is null:

- If $y > 0$, then $\delta_0(y) = 0$ and $f_{ZINB}(y;\mu,\theta,\pi) = (1-\pi)f_{NB}(y;\mu,\theta)$, so we obtain:

$$
\begin{aligned}
\frac{\partial}{\partial\mu}\ln f_{ZINB}(y;\mu,\theta,\pi) &= \frac{y}{\mu} - \frac{y+\theta}{\mu+\theta}, \\
\frac{\partial}{\partial\pi}\ln f_{ZINB}(y;\mu,\theta,\pi) &= \frac{-1}{1-\pi}.
\end{aligned}
$$

- If $y = 0$, then $\delta_0(y) = 1$, and we get

$$
\begin{aligned}
\frac{\partial}{\partial\mu}\ln f_{ZINB}(0;\mu,\theta,\pi) &= \frac{-(1-\pi)\left(\frac{\theta}{\mu+\theta}\right)^{\theta+1}}{\pi+(1-\pi)\left(\frac{\theta}{\mu+\theta}\right)^\theta}, \\
\frac{\partial}{\partial\pi}\ln f_{ZINB}(0;\mu,\theta,\pi) &= \frac{1-\left(\frac{\theta}{\mu+\theta}\right)^\theta}{\pi+(1-\pi)\left(\frac{\theta}{\mu+\theta}\right)^\theta}.
\end{aligned}
$$

When $N \geq 1$, using standard calculus for the differentiation of compositions and the facts that:

$$
\begin{aligned}
\left(\ln^{-1}\right)'(\ln\mu) &= \mu, \\
\left(\mathrm{logit}^{-1}\right)'(\mathrm{logit}\pi) &= \pi(1-\pi),
\end{aligned}
$$

we finally get that

$$
\begin{aligned}
\nabla_{a_\mu}F &= A_\mu^\top G, \\
\nabla_{a_\pi}F &= A_\pi^\top H, \\
\nabla_b F &= B_\mu^\top G + B_\pi^\top H.
\end{aligned}
$$

where $G$ and $H$ are the $N$-dimensional vectors given by

$$
\begin{aligned}
\forall i \in [1,N], \quad G_i &= \mu_i\frac{\partial}{\partial\mu}\ln f_{ZINB}(y_i;\mu_i,\theta_i,\pi_i), \\
H_i &= \pi_i(1-\pi_i)\frac{\partial}{\partial\pi}\ln f_{ZINB}(y_i;\mu_i,\theta_i,\pi_i).
\end{aligned}
$$

**Simulated data sets.** Simulating from the ZINB-WaVE model: In order to simulate realistic data, we fitted our ZINB-WaVE model to two real data sets (V1 and S1/CA1) and used the resulting parameter estimates as the truth to be estimated in the simulation. Genes that did not have at least five reads in at least five cells were filtered out and $J = 1000$ genes were then sampled at random for each data set. The ZINB-WaVE model was fit to the count matrix $Y$ with the number of unknown cell-level covariates set to $K = 2$, genewise dispersion ($\zeta = \ln\theta = -\ln\phi$), $X_\mu$, $X_\pi$, $V_\mu$, and $V_\pi$ as columns of ones (i.e., intercept only), and no offset matrices, to get estimates for $W$, $\alpha_\mu$, $\alpha_\pi$, $\beta_\mu$, $\beta_\pi$, $\gamma_\mu$, $\gamma_\pi$, and $\zeta$. The parameters which were varied in the simulations are the number of cells, the proportion of zero counts, and the ratio of within to between-cluster sums of squares for $W$. Details on the parameter choices follow.

The number of cells was set to $n = 100; 1000; 10,000$.

The proportion of zero counts, $zfrac = \sum 1(Y_{ij} = 0)/nJ$, was set via the parameter $\gamma_\pi$: $zfrac \approx 0.25, 0.50, 0.75$. As $\mathrm{logit}(\pi) = X\beta_\pi + (V\gamma_\pi)^\top + W\alpha_\pi$, the value of $\gamma_\pi$ is directly linked to the dropout probability $\pi$, thus to the zero fraction. Note that by changing only $\gamma_\pi$ but not $\gamma_\mu$, we change the dropout rate but not the underlying, unobserved mean expression, i.e., this varies the number of *technical* zeros but not *biological* zeros.

The ratio of within to between-cluster sums of squares for $W$ was set in the following way. Let $C$ denote the number of clusters and $n_c$ the number of cells in cluster $c$. For a given column of $W$ (out of $K$ columns), let $W_{ic}$ denote the value for cell $i$ in cluster $c$, $\overline{W}$ the overall average across all $n$ cells, $\overline{W}_c$ the average for cells in cluster $c$, and TSS the total sum of squares. Then,

$$
\begin{aligned}
\mathrm{TSS} &= \sum_{c=1}^C \sum_{i=1}^{n_c}\left(W_{ic} - \overline{W}\right)^2 \\
&= \sum_{c=1}^C \sum_{i=1}^{n_c}\left(W_{ic} - \overline{W}_c\right)^2 + \sum_{c=1}^C n_c\left(\overline{W}_c - \overline{W}\right)^2 \\
&= \mathrm{WSS} + \mathrm{BSS},
\end{aligned}
$$

with $\mathrm{WSS} = \sum_{c=1}^C \sum_{i=1}^{n_c}\left(W_{ic} - \overline{W}_c\right)^2$ and $\mathrm{BSS} = \sum_{c=1}^C n_c\left(\overline{W}_c - \overline{W}\right)^2$ the within and between-cluster sums of squares, respectively. The level of difficulty of the clustering problem can be controlled by the ratio of within to between-cluster sums of squares. However, we want to keep the overall mean $\overline{W}$ and overall variance (i.e., TSS) constant, so that the simulated values of $W$ stay in the same range as the estimated $W$ from the real data set; this prevents us from simulating an unrealistic count matrix $Y$.

Let us scale the between-cluster sum of squares by $a^2$ and the within-cluster sum of squares by $a^2 b^2$, i.e., replace $\left(\overline{W}_c - \overline{W}\right)$ by $a\left(\overline{W}_c - \overline{W}\right)$ and $\left(W_{ic} - \overline{W}_c\right)$ by $ab\left(W_{ic} - \overline{W}_c\right)$, with $a \geq 0$ and $b \geq 0$ such that $TSS$ and $\overline{W}$ are constant. The total sum of squares $TSS$ remains constant, i.e.,

$$
\mathrm{TSS} = a^2 b^2 \mathrm{WSS} + a^2 \mathrm{BSS},
$$

provided

$$
a^2 = \frac{\mathrm{TSS}}{b^2 \mathrm{WSS} + \mathrm{BSS}}.
$$

Requiring the overall mean $\overline{W}$ to remain constant implies that

$$\overline{W}_c^* = (1-a)\overline{W} + a\overline{W}_c,$$

where the * superscript refers to the transformed $W$. Thus,

$$
\begin{aligned}
W_{ic}^* &= \overline{W}_c^* + ab(W_{ic} - \overline{W}_c) \\
&= (1-a)\overline{W} + a\overline{W}_c + ab(W_{ic} - \overline{W}_c) \\
&= (1-a)\overline{W} + a(1-b)\overline{W}_c + abW_{ic}.
\end{aligned}
\tag{11}
$$

The above transformation results in a scaling of the ratio of within to between-cluster sums of squares by $b^2$, while keeping the overall mean and variance constant.

In our simulations, we fixed $C = 3$ clusters and considered three values for $b^2$, where the same value of $b^2$ is applied to each of the $K$ columns of $W$: $b^2 = 1$, corresponding to the case where the within and between-cluster sums of squares are the same as the ones of the fitted $W$ from the real data set; $b^2 = 5$, corresponding to a larger ratio of within to between-cluster sums of squares and hence a harder clustering problem; $b^2 = 10$, corresponding to a very large ratio of within to between-cluster sums of squares and hence almost no clustering.

Overall, 2 (real data sets)) $\times$ 3 ($n$) $\times$ 3 (zfrac)) $\times$ 3 (clustering) = 54 scenarios were considered in the simulation.

For each of the 54 scenarios, we simulated $B = 10$ data sets, resulting in a total of $54 \times 10 = 540$ data sets. Using the fitted $W$, $\alpha_\mu$, $\alpha_\pi$, $\beta_\mu$, $\beta_\pi$, $\gamma_\mu$, $\gamma_\pi$, and $\zeta$ from one of the two real data sets, the data sets were simulated according to the following steps.

1. Simulate $W$ with desired clustering strength. First fit a $K$-variate Gaussian mixture distribution to $W$ inferred from one of the real data sets using the R function `Mclust` from the mclust package and specifying the number of clusters $C$. Then, for each of $B = 10$ data sets, simulate $W$ cluster by cluster from $K$-variate Gaussian distributions using the `mvrnorm` function from the MASS package, with the cluster means, covariance matrices, and frequencies output by `Mclust`. Transform $W$ as in Eq. (11) to get the desired ratio of within to between-cluster sums of squares.
2. Simulate $\gamma_\mu$ and $\gamma_\pi$ to get the desired zero fraction. We only considered cell-level intercept $n$-vectors $\gamma_\mu$ and $\gamma_\pi$, i.e., $L = 1$ and a matrix $V$ of gene-level covariates consisting of a single column of ones. As the fitted $\gamma_\mu$ and $\gamma_\pi$ from the original data sets are correlated $n$-vectors, fit a bivariate Gaussian distribution to $\gamma_\mu$ and $\gamma_\pi$ using the function `Mclust` from the mclust package with $C = 1$ cluster. Then, for each of $B = 10$ data sets, simulate $\gamma_\mu$ and $\gamma_\pi$ from a bivariate Gaussian distribution using the `mvrnorm` function from the MASS package, with the mean and covariance matrix output by `Mclust`. To increase/decrease the zero fraction, increase/decrease each element of the mean vector for $\gamma_\pi$ inferred from `Mclust` (shifts of {0,2,5} for V1 data set and {−1.5, 0.5, 2} for S1/CA1 data set).
3. Create the ZINB-WaVE model using the function `zinbModel` from the package zinbwave.
4. Simulate counts using the function `zinbSim` from the package zinbwave.

Simulating from the Lun & Marioni[42] model: To simulate data sets from a different model than our ZINB-WaVE model, we simulated counts using the procedure described in Lun & Marioni[42] (details in Supplementary Materials of original publication and code available from the Github repository https://github.com/MarioniLab/PlateEffects2016). Although the Lun & Marioni[42] model is also based on a ZINB distribution, the distribution is parameterized differently and also fit differently, gene by gene. In particular, the negative binomial mean is parameterized as a product of the expression level of interest and two nuisance technical effects, a gene-level effect assumed to have a log-normal distribution and a cell-level effect (cf. library size) whose distribution is empirically derived. The zero-inflation probability is assumed to be constant across cells for each gene and is estimated independently of the negative binomial mean. The ZINB distribution is fit gene by gene using the `zeroinfl` function from the R package pscl. We used the raw gene-level read counts for the mESC data set as input to the script `reference/submitter.sh`, to create a simulation function constructed by fitting a ZINB distribution to these counts. The script `simulations/submitter.sh` was then run to simulate counts based on the estimated parameters (negative binomial mean, zero inflation probability, and genewise dispersion parameter). We simulated $C = 3$ clusters with equal number of cells per cluster. The parameters which were varied in the simulations are as follows.

The number of cells was set to $n = 100$; 1000; 10,000.

The proportion of zero counts, zfrac $= \sum_{i,j} 1(Y_{ij} = 0)/nJ$, was set via the zero inflation probability: zfrac $\approx 0.4$, 0.6, 0.8. For zfrac $= 0.4$, we did not modify the code in Lun & Marioni[42]. However, to simulate data sets with greater zero fractions, namely zfrac $= 0.6$ and zfrac $= 0.8$, we added respectively 0.3 and 0.6 to the zero-inflation probability ($p_i'$, in their notation).

For each of the 3 ($n$) $\times$ 3 (zfrac) = 9 scenarios, we simulated $B$ ($B = 10$) data sets, resulting in 9) $\times$ 10 = 90 simulated data sets in total.

**Dimensionality reduction methods.** Three different dimensionality reduction methods were applied to the real and simulated data sets: ZINB-WaVE, zero-inflated factor analysis, and PCA. For all the methods, we selected $K = 2$

dimensions, unless specified otherwise. A notable exception is the S1/CA1 data set, for which, given the large number of cells and the complexity of the signal, we specified $K = 3$ dimensions.

ZINB-WaVE: We applied the ZINB-WaVE procedure using the function zinbFit from our R package zinbwave, with the following parameter choices. Number of unknown cell-level covariates $K$: $K = 1, 2, 3, 4$. Gene-level covariate matrix $V$: not included or set to a column of ones $1_J$. Cell-level covariate matrix $X$: set to a column of ones $1_n$. For the mESC data set, we also considered including batch covariates in $X$. Dispersion parameter $\zeta$: the same for all genes (common dispersion) or specific to each gene (genewise dispersion).

Zero-inflated factor analysis: We used the zero-inflated factor analysis (ZIFA) method[26], as implemented in the ZIFA python package (Version 0.1) available at https://github.com/epierson9/ZIFA, with the block algorithm (function block_ZIFA.fitModel, with default parameters). The output of ZIFA is an $n \times K$ matrix corresponding to a projection of the counts onto a latent low-dimensional space of dimension $K$.

PCA: We used the function prcomp from the R package stats for the simulation study and, for computational efficiency, the function jsvds from the R package rARPACK for the real data sets.

**Normalization methods.** As normalization is essential, especially for zero-inflated distributions, PCA and ZIFA were applied to both raw and normalized counts. The following normalization methods were used.

Total count normalization (TC): Counts are divided by the total number of reads sequenced for each sample and multiplied by the mean total count across all the samples. This method is related to the popular transcripts per million (TPM)[52] and fragments per kilobase million (FPKM)[53] methods.

Full-quantile normalization (FQ)[54]: The quantiles of the distributions of the gene-level read counts are matched across samples. We used the function between Lane Normalization from the Bioconductor R package EDASeq.

Trimmed mean of M values (TMM)[55]: The TMM global-scaling factor is computed as the weighted mean of log-ratios between each sample and a reference sample. If the majority of the genes are not differentially expressed (DE), TMM should be close to 1, otherwise, it provides an estimate of the correction factor that must be applied in order to fulfill this hypothesis. We used the function calcNormFactors from the Bioconductor R package edgeR to compute these scaling factors.

**Clustering methods.** Clustering of the OE data set: We used the resampling-based sequential ensemble clustering (RSEC) framework implemented in the RSEC function from the Bioconductor R package clusterExperiment[56]. Briefly, RSEC implements a consensus clustering algorithm which generates and aggregates a collection of clusterings, based on resampling cells and using a sequential tight clustering algorithm[57]. See Fletcher et al.[29] for details on the parameters used for RSEC in the original analysis and Perraudeau et al.[31] for details on the parameters used for RSEC in the ZINB-WaVE workflow.

Clustering of the 10$\times$ Genomics 68k PBMCs data set: We used two different clustering methods to assess the ability of ZINB-WaVE to extract biologically meaningful signal from the data.

First, we used a a clustering procedure similar to the one implemented in the R package Seurat[34] (Version 2.0.1). In particular, we used ZINB-WaVE ($K = 10$) instead of PCA for dimensionality reduction. The clustering is based on shared nearest neighbor modularity[33]. We used the following parameters: `k.param=10`, `k.scale=10`, `resolution=0.6`. All the other parameters were left at their default values.

We also clustered the data using a sequential $k$-means clustering approach[57], implemented in the `clusterSingle` function of the clusterExperiment package[56]. We used the following parameters: `sequential=TRUE`, `subsample=FALSE`, `k0=15`, `beta=0.95`, `clusterFunction="kmeans"`.

**Models for count data.** We compared the goodness-of-fit of our ZINB-WaVE model to that of two other models for count data: a standard negative binomial model, that does not account for zero inflation, and the MAST hurdle model, that is specifically designed for scRNA-seq data[23].

Goodness-of-fit was assessed on the V1 data set using mean-difference plots (MD-plots) of estimated vs. observed mean count and zero probability, as well as plots of the estimated dispersion parameter against the observed zero frequency.

ZINB-WaVE model: For our ZINB-WaVE model, the overall mean and zero probability are

$$E[Y_{ij}] = (1 - \pi_{ij})\mu_{ij},$$

$$P(Y_{ij} = 0) = \pi_{ij} + (1 - \pi_{ij})\left(1 + \phi_j \mu_{ij}\right)^{\frac{1}{\phi_j}}.$$

ZINB-WaVE was fit to the V1 data set using the function `zinbFit` from our R package zinbwave, with the following parameter choices: $K = 0$ unknown cell-level covariates, gene-level intercept ($X = 1_n$), cell-level intercept ($V = 1_J$), and genewise dispersion.

Negative binomial model: The negative binomial (NB) distribution is a special case of the ZINB distribution that does not account for zero inflation, i.e., for which $\pi = 0$. Thus,

$$
\begin{aligned}
E[Y_{ij}] &= \mu_{ij}, \\
P(Y_{ij} = 0) &= \left(1 + \phi_j \mu_{ij}\right)^{\frac{1}{\phi_j}}.
\end{aligned}
$$

A NB distribution was fit gene by gene to the V1 data set, after full-quantile normalization, using the Bioconductor R package edgeR[41] (Version 3.16.5) with only an intercept (i.e., default value for the `design` argument as a single column of ones) and genewise dispersion.

Model-based analysis of single-cell transcriptomics: The model-based analysis of single-cell transcriptomics (MAST) hurdle model proposed by Finak et al.[23] is defined as follows:

$$
\begin{aligned}
\text{logit}(P(Z_{ij} = 1)) &= X_i \beta_j^D, \\
Y_{ij} | Z_{ij} = 1 &\sim \mathcal{N}\left(X_i \beta_j^C, \sigma_j^2\right),
\end{aligned}
$$

where $Y_{ij}$ is log2(TPM +1) for cell $i$ and gene $j$, $X_i \in \mathbb{R}^k$ is a known covariate vector, and $Z_{ij}$ indicates whether gene $j$ is truly expressed in cell $i$. For each gene $j$, the parameters of the MAST model are: the regression coefficients $\beta_j^D \in \mathbb{R}^k$ for the discrete part and the regression coefficients $\beta_j^C \in \mathbb{R}^k$ and variance $\sigma_j^2 \in \mathbb{R}$ for the Gaussian continuous part. Note that we follow the notation in Finak et al.[23], which is different from that in the ZINB-WaVE model of Eq. (4).

For the MAST model, the overall mean and zero probability are

$$
\begin{aligned}
E[Y_{ij}] &= \text{logit}^{-1}\left(X_i \beta_j^D\right) X_i \beta_j^C, \\
P(Y_{ij} = 0) &= 1 - \text{logit}^{-1}\left(X_i \beta_j^D\right).
\end{aligned}
$$

MAST is implemented in the Bioconductor R package MAST[58] (which allows different covariate vectors $X_i$ for the continuous and discrete components). We use the function `zlm` to fit MAST with an intercept and a covariate for the cellular detection rate (as recommended in the MAST vignette for the MAIT data analysis) for both the discrete and continuous parts.

Note that in contrast to the NB and ZINB models, the MAST hurdle model is for log2(TPM+1) instead of counts and has no dispersion parameter. To be able to compare the fits of the three models, the MAST goodness-of-fit plots display estimated vs. observed mean log2(TPM+1) and estimated variance $\sigma_j^2$ vs. observed zero frequency (Supplementary Fig. 21). Although not a direct comparison, we think this allows a fair assessment of the goodness-of-fit of the different models.

**Evaluation criteria.** Clustering: Silhouette width: Given a set of labels for the cells (e.g., biological condition, batch), silhouette widths provide a measure of goodness of the clustering of the cells with respect to these labels. Silhouette widths may be averaged within clusters or across all observations to assess clustering strength at the level of clusters or overall, respectively. The silhouette width $s_i$ of a sample $i$ is defined as follows:

$$
s_i = \frac{b_i - a_i}{\max\{a_i, b_i\}},
$$

where $a_i = d\left(i, \mathcal{C}_{cl(i)}\right)$, $b_i = \min_{l \neq cl(i)} d(i, \mathcal{C}_l)$, $\mathcal{C}_{cl(i)}$ is the cluster to which $i$ belongs, and $d(i, \mathcal{C}_l)$ is the average distance between sample $i$ and the samples in cluster $\mathcal{C}_l$.

Average silhouette widths were used to compare ZINB-WaVE, PCA, and ZIFA on both real and simulated data sets. For simulated data, the cluster labels correspond to the true simulated $W$ and, for each scenario, silhouette widths were computed and averaged over $B$ data sets. For real data, the authors' cluster labels or known cell types were used.

Clustering: Precision and recall: Two different clusterings (i.e., partitions) may be compared quantitatively using the precision and recall coefficients. These measures involve assessing whether pairs of cells cluster together in each of the two clusterings. Let YY (respectively NY; YN; and NN) be the total number of pairs of cells which are in the same cluster in both Clustering 1 and Clustering 2 (respectively, in different clusters in Clustering 1 but in the same cluster in Clustering 2; in the same cluster in Clustering 1 but in different clusters in Clustering 2; in different clusters in both Clustering 1 and Clustering 2). Then, using Clustering 1 as a reference, the precision coefficient is defined as the proportion of cells clustered together in Clustering 2 which are also (correctly) clustered together in Clustering 1

$$
\text{Precision} = \frac{\text{YY}}{\text{YY} + \text{NY}}.
$$

Similarly, the recall coefficient is defined as the proportion of cells (correctly) clustered together in Clustering 1 which are also clustered together in Clustering 2

$$
\text{Recall} = \frac{\text{YY}}{\text{YY} + \text{YN}}.
$$

Precision and recall were used to compare ZINB-WaVE, PCA, and ZIFA on simulated data sets, where the reference Clustering 1 corresponds to the true simulated labels and Clustering 2 was performed using the R function `kmeans` in the reduced-dimensional space: the first two columns of $W$ for ZINB-WaVE, the first two principal components for PCA, and the first two latent variables for ZIFA. We simulated 3 clusters and set the number of clusters equal to 3 in `kmeans` (argument `centers`). Precision and recall were computed using the `extCriteria` function from the R package clusterCrit[59].

Dimensionality reduction: Correlation with QC measures: To evaluate the dependence of the inferred low-dimensional signal on unwanted variation, we computed the absolute correlation between each dimension (e.g., principal component) and a set of quality control (QC) measures.

For the V1 data set, FastQC (http://www.bioinformatics.babraham.ac.uk/projects/fastqc/) and Picard tools (https://broadinstitute.github.io/picard/) were used to compute a set of 16 QC measures. These measures are available as part of the Bioconductor R package scRNAseq (https://bioconductor.org/packages/scRNAseq). For the glioblastoma and mESC data sets, we used the scater Bioconductor R package[60] (Version 1.2.0) to compute a set of 7 QC measures. For the S1/CA1 data set, we used a set of 6 QC measures provided by the authors (http://linnarssonlab.org/cortex/).

Dimensionality reduction: Correlation of pairwise distances between observations: For simulated data, we assessed different dimensionality reduction methods (ZINB-WaVE, PCA, and ZIFA) in terms of the correlation between pairwise distances between observations in the true and in the estimated reduced-dimensional space. In particular, we monitored the influence of the number of unknown cell-level covariates $K$ when the other parameters are set correctly ($V$ is a column of ones $\mathbf{1}_J$ and genewise dispersion). For each simulation scenario, the correlation between true and estimated pairwise distances was computed and averaged over $B$ data sets.

Bias and MSE of the ZINB-WaVE estimators: For data simulated from the ZINB-WaVE model, let $\theta$ and $\hat{\theta}_b$, $b = 1, \ldots, B$, respectively denote the true parameter and an estimator of this parameter for the $b$th simulated data set. Then, for each scenario, the performance of the estimator $\hat{\theta}$ can be assessed in terms of bias and mean squared error (MSE) as follows:

$$
\begin{aligned}
\text{Bias} &= \frac{1}{B} \sum_{b=1}^{B} \left(\hat{\theta}_b - \theta\right), \\
\text{MSE} &= \frac{1}{B} \sum_{b=1}^{B} \left(\hat{\theta}_b - \theta\right)^2.
\end{aligned}
$$

Bias and MSE were computed for $\beta_\mu$, $\beta_\pi$, $\gamma_\mu$, $\gamma_\pi$, $\zeta$, $W\alpha_\mu$, $W\alpha_\pi$, $\ln(\mu)$, and $\text{logit}(\pi)$. When the parameter to be estimated was an $n \times J$ matrix, the matrix was converted to a $1 \times nJ$ row vector and bias and MSE were averaged over the elements of the vector.

Model selection for ZINB-WaVE: The Akaike information criterion (AIC) and the Bayesian information criterion (BIC) are widely used for model selection and are defined as follows:

$$
\begin{aligned}
\text{AIC} &= 2N - 2\ell\left(\hat{\beta}, \hat{\gamma}, \hat{W}, \hat{\alpha}, \hat{\zeta}\right), \\
\text{BIC} &= \ln(n)N - 2\ell\left(\hat{\beta}, \hat{\gamma}, \hat{W}, \hat{\alpha}, \hat{\zeta}\right),
\end{aligned}
$$

where $\ell\left(\hat{\beta}, \hat{\gamma}, \hat{W}, \hat{\alpha}, \hat{\zeta}\right)$ is the log-likelihood function evaluated at the MLE (Eq. (5)), $n$ is the sample size, and $N$ is the total number of estimated parameters, i.e., $N = J(M_\mu + M_\pi) + n(L_\mu + L_\pi) + 2KJ + nK + J$ when the model is fit with genewise dispersion and $N = J(M_\mu + M_\pi) + n(L_\mu + L_\pi) + 2KJ + nK + 1$ when the model is fit with common dispersion.

We use AIC and BIC in the simulation to select the number of unknown cell-level covariates ($K$).

Note that because of the complex non-convex likelihood function for the ZINB-WaVE model, there is no closed-form expression for the MLE and our numerical optimization procedure only provides an approximation of the AIC and BIC. In practice, however, we found that our results closely approximated the true MLE (see Results and Supplementary Fig. 22).

**Real data sets.** V1 data set: Tasic et al.[3] characterized more than 1600 cells from the primary visual cortex (V1) in adult male mice, using a set of established Cre lines. Single cells were isolated by FACS into 96-well plates and RNA was reverse transcribed and amplified using the SMARTer kit. Sequencing was performed using the Illumina HiSeq platform, yielding 100 bp-long reads. We selected a subset of three Cre lines, Ntsr1-Cre, Rbp4-Cre, and Scnn1a-Tg3-Cre, that label layer 4,

layer 5, and layer 6 excitatory neurons, respectively. This subset consists of 379 cells, grouped by the authors into 17 clusters; we excluded the cells that did not pass the authors' quality control filters and that were classified by the authors as "intermediate" cells between two clusters, retaining a total of 285 cells. Gene expression was quantified by gene-level read counts. Raw gene-level read counts and QC metrics (see below) are available as part of the scRNAseq Bioconductor R package (https://bioconductor.org/packages/scRNAseq). We applied the dimensionality reduction methods to the 1000 most variable genes.

S1/CA1 data set: Zeisel et al.[4] characterized 3005 cells from the primary somatosensory cortex (S1) and the hippocampal CA1 region, using the Fluidigm C1 microfluidics cell capture platform followed by Illumina sequencing. Gene expression was quantified by UMI counts. In addition to gene expression measures, we have access to metadata that can be used to assess the methods: batch, sex, number of mRNA molecules. Raw UMI counts and metadata were downloaded from http://linnarssonlab.org/cortex/.

mESC data set: Kolodziejczyk et al.[37] sequenced the transcriptome of 704 mouse embryonic stem cells (mESCs), across three culture conditions (serum, 2i, and a2i), using the Fluidigm C1 microfluidics cell capture platform followed by Illumina sequencing. We selected only the cells from the second and third batch, after excluding the samples that did not pass the authors' QC filtering. This allowed us to have cells from each culture condition in each batch and resulted in a total of 169 serum cells, 141 2i cells, and 159 a2i cells. In addition to gene expression measures, we have access to batch and plate information that can be included as covariates in our model. Raw gene-level read counts were downloaded from http://www.ebi.ac.uk/teichmann-srv/espresso/. Batch and plate information was extracted from the sample names, as done in Lun & Marioni[42]. We applied the dimensionality reduction methods to the 1000 most variable genes.

Glioblastoma data set: Patel et al.[6] collected 672 cells from five dissociated human glioblastomas. Transcriptional profiles were generated using the SMART-Seq protocol. We analyzed only the cells that passed the authors' QC filtering. The raw data were downloaded from the NCBI GEO database (accession GSE57872). Reads were aligned using TopHat −rg-library Illumina −rg-platform Illumina −keep-fasta-order -G -N 3 −read-edit-dist 3 −no-coverage-search -x 1 -M -p 12. Counts were obtained using `htseq-count` with the following parameters (http://www-huber.embl.de/HTSeq/doc/count.html): -a 10 -q -s no -m union. We applied the dimensionality reduction methods to the 1000 most variable genes.

OE data set: Fletcher et al.[29] characterized 849 FACS-purified cells from the mouse OE, using the Fluidigm C1 microfluidics cell capture platform followed by Illumina sequencing. Gene-level read counts were downloaded from GEO (GSE95601; file `GSE95601_oeHBCdiff_Cufflinks_eSet_counts_table.txt.gz`). As done in Perraudeau et al.[31], we filtered the cells that exhibited poor sample quality using SCONE[36] (v. 1.1.2). A total of 747 cells passed this filtering procedure. To compare with the original results, we also re-analyze the final repertoire of 13 stable clusters found in Fletcher et al.[29], consisting of 616 cells, downloaded from https://github.com/rufletch/p63-HBC-diff. See Fletcher et al.[29] for details on the original analysis and Perraudeau et al.[31] for details on the ZINB-WaVE based workflow.

10× Genomics 68k PBMCs data set: Zheng et al.[32] characterized 68,579 peripheral blood mononuclear cells (PBMCs) from a healthy donor. The gene-level UMI counts were downloaded from https://www.10xgenomics.com/single-cell/ using the cellrangerRkit R package (Version 1.1.0). We applied the dimensionality reduction methods to the 1000 most variable genes.

**Data availability**. The ZINB-WaVE method is implemented in the open-source R package zinbwave, available as part of the Bioconductor Project (https://bioconductor.org/packages/zinbwave). See the package vignette for a detailed example of a typical use. The code to reproduce all the analyses and figures of this article is available at https://github.com/drisso/zinb_analysis.

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

## Acknowledgements

We would like to thank Olivier Mirabeau from the Curie Institute for the pre-processing of the Glioblastoma data set. We also thank Aaron Lun and the Marioni Lab for making the code needed for the simulations available online. Finally, we are grateful to Russell Fletcher for his help with the re-analysis of the OE data. D.R. and S.D. are supported by the National Institutes of Health BRAIN Initiative (Grant U01 MH105979, PI: John Ngai). S.G. and J.-P.V. are supported by the French National Research Agency (Grant ABS4NGS ANR-11-BINF-0001). J.-P.V. is supported by the European Research Council (Grant ERC-SMAC-280032), the Miller Institute for Basic Research in Science, and the Fulbright Foundation.

## Author contributions

D.R., S.D., and J.-P.V. formulated the statistical model. S.G. and J.-P.V. conceived and implemented the optimization algorithm. D.R., S.G., F.P., and J.-P.V. wrote the R package. D.R. performed the real data analysis. D.R., F.P., S.D., and J.-P.V. designed the simulation study. F.P. performed the simulations and analyzed the simulated data. D.R., F.P., S.D., and J.-P.V. wrote the manuscript.

## Additional information

**Competing interests:** The authors declare no competing financial interests.

