## [Peer Review File · Nature Communications]

Reviewers' Comments:

Reviewer #1:

Remarks to the Author:

I think this is a nice paper that presents a statistically sound and novel method for the analysis of single-cell RNA-seq data. Overall, I like the approach, and I like that the authors have developed a model that truly models the bimodality of the data for dimension reduction. The regression framework is also very relevant for adjusting for biological and technical covariates. In my opinion, it would be a nice addition to Nature Communications. I have several comments/questions listed below, that I think should be useful to the authors to improve their paper and hopefully increase the overall scientific impact.

Major comments:

Biological relevance: While I really appreciate the amount of work that was put into the different comparisons (silhouette scores across methods and normalizations). I would have liked to see an application of ZINB-WAVE that leads to new biological insight, or at least some sort of new application. For example, it would be interesting to see how this method can be used to identify new cell subsets or find novel structure in the data (particularly after batch correction). Or as an alternative, show that some new structures identified in published datasets were likely driven by batch effects that were not accounted for. I am sure there are plenty of those out there!

Datasets: Most of the dataset used are fairly low throughput. Given the goal of the proposed method (dimension reduction), I think it would be good to try to use a larger dataset with tens of thousands of cells. I suggest the use of the 10X dataset generated by the Bielas lab (<https://www.nature.com/articles/ncomms14049>). It is a large dataset with PBMCs, and sorted cell populations. I think it would be very useful for motivating and validating ZINB-WAVE. This might also help ZINB-WAVE as these data are very sparse with a lot of zero.

Comparison to t-SNE: It seems that t-SNE really is the best current practice for dimension reduction and visualization of single-cell data. Given this, I think it would be useful to include t-SNE in your comparison. I am personally not a big fan of t-SNE, and I know that since it's not linear it might not be a fair comparison, but nevertheless I think it would be a useful comparison for the average reader.

Minor comments:

Although the main goal of the proposed approach is dimension reduction, the zero-inflated model shares some similarities with MAST. ZINB-WAVE is probably better in terms of goodness-of-fit, but it would be nice to show that.

Number of latent factors. Do you provide any guidance on how many factors should be included in the model? It seems that you only used two for the results presented, is that correct?

Down-selection of genes? Is the proposed approach sensitive to the number of genes used? Why use the 1000 most variable genes, and not the top 500 or 100? How does the approach scale with the number of genes used? How fast is the approach?

Wanted vs Unwanted variation: Is there a way to know if some of the variation observed in your lower dimensional space is biologically relevant or unwanted (due to nuisance factor). I understand this is difficult to do, but it would be nice to discuss this and perhaps provide some intuitions. Perhaps the first recommendation would be to see if cells cluster according to some technical variable (plate, date, etc). Again, this might be useful to the reader.

In the introduction, you say “many genes fail to be detected even though they are expressed”. I think it would be important to say that many genes fail to be detected because they are not expressed, so it might be good to clarify this.

In the introduction, you say ““bulk” RNA sequencing, which is only capable of measuring gene expression levels averaged over millions of cells”. This was true 10 years ago, but today one can easily do “bulk” RNA-sequencing on small cell subsets with thousands of cells and possibly dozens of cells if one uses an optimized protocol (e.g. SMART-seq). So I suggest that you modify this statement.

Reviewer #2:

Remarks to the Author:

This manuscript describes a new method ZINB-WaVE for analyzing single-cell RNA-seq (scRNA-seq) data. scRNA-seq is an emerging technology for measuring the transcriptome of individual cells. It provides a powerful tool to study cell-to-cell heterogeneity. The data from scRNA-seq experiments contain multiple sources of technical biases and noises. Computational and statistical methods play crucial roles in effectively using such data. This manuscript represents a timely study trying to address an important problem in scRNA-seq analysis. It attempts to represent scRNA-seq data using a low-dimensional structure while accounting for dropouts. The ZINB-WaVE method extends the RUV approach developed previously by the same group to zero-inflated negative binomial count data. It allows one to include sample-level or gene-level covariates in the model. Through simulations and real data, the authors demonstrate that ZINB-WaVE performs better than other dimension reduction methods including PCA and ZIFA. Overall, the manuscript is clearly written, and the proposed method is useful for single-cell RNA-seq data analysis. I do, however, have several questions that I hope the authors can answer.

Major comments:

1. A nice feature of ZINB-WaVE is its ability to handle gene-level covariates such as GC content and gene length. This is also a major difference between ZINB-WaVE and existing methods such as PCA and ZIFA. However, this important feature and its practical value have not been demonstrated anywhere in the manuscript. It will be very useful if the authors can demonstrate the use of this important feature in real applications.
2. In the ZINB-WaVE model, the offset matrices O_{μ} and O_{π} in formulas (4) and (5) are assumed to be known. Each of these matrices contains $n \times J$ entries. That is a large number of parameters. It is unclear to me why these two offset matrices are known and how they are specified?
3. The ZINB-WaVE procedure also assumes that the number of unknown factors (i.e., unknown cell-level covariates) K is given. However, in practice K is unknown. The present manuscript does not provide any procedure to choose K . Although the authors argue that their method is relatively robust to misspecified K , Figure 5 and other related figures show that the choice of K does influence the bias and MSE. Thus, it is important to provide a method that can help users to objectively choose the optimal K .
4. When comparing ZINB-WaVE with other methods, the data shown in the scatterplots may be misleading. In Figure 2, Figure S4, and Figure S5, the scatterplots for PCA and ZIFA are not based on

their best performance. The authors should show the plots from the best normalization method (FQ). The authors showed how the normalization method affects the dimension reduction results of PCA in Figure S8. It will be interesting to see how it affects the results in all 4 real datasets for both PCA and ZIFA.

5. The authors showed that their method can be used to adjust for batch effects. To more convincingly demonstrate this advantage, they should compare ZINB-WaVE with PCA and ZIFA in the real data analysis. For example, what if one first uses existing batch effect correction methods such as ComBat to adjust for batch effect and then runs PCA and ZIFA? How does this approach perform compared to ZINB-WaVE?

6. The authors used the average silhouette width to evaluate the clustering performance of each method. In real applications, the true clustering structure is unknown. Therefore, it will be useful if the authors also evaluate different methods by first performing clustering based on the low-dimensional signals extracted by different methods and then evaluate how accurate the clustering analysis recovers the true cell clusters (identities).

Minor comments:

7. Page 4, paragraph 2, line 2: "and the second component was highly correlated with dropout and detection rates (Fig. 2d)". Here "the second component" in Fig 2d does not correlate highly with dropout. The "first" component in Fig 2d does.

8. Page 4, paragraph 3: "These observations are not limited to the Glioblastoma dataset, and the same trend was observed for ... S1/CA1 dataset (Supplementary Fig. S5), ...". This statement is inconsistent with Supplementary Fig. S5 in which the first dimension of ZINB-WaVE showed higher correlation with coverage and detection rate than ZIFA, and higher detection rate than PCA.

9. Page 15, section "Initialization", step 3: "Set $Z_{ij}=1$ if (i,j) belongs to P ". Here if (i,j) belongs to P , Z_{ij} should be 0 instead of 1 according to the definition of Z_{ij} .

Reviewer #3:

Risso et al. propose a new method, ZinbWave, that extends the RUV model to account for noise models that are specific to scRNA-seq. They test their model on simulated and real data, benchmarking against PCA and ZIFA.

From a statistical standpoint, I enjoyed reading the manuscript. It lays out the problem clearly, and the ZINB-WAVE model does account for the vast majority of (at least known) sources of noise for scRNA-seq. Specific features I appreciated included :

1. The use of both sample and gene-level covariates. Most models do not consider the possibility of gene-level covariates, but this is an interesting addition (however there is no analysis shown on how much these improve the results, which is a shame)

2. Since the model includes size factors, it can run on non-normalized counts. Therefore, the procedure handles normalization and dimensional reduction in a single integrated step, and I believe this is well-motivated and a desired property for scRNA-seq workflows.

3. The optimization and learning procedure is non-trivial, to say the least, though there is always some concern that the procedure could converge to a local minima, which is not a concern for PCA.

Therefore, I believe the manuscript represents a theoretical advance for the analysis of scRNA-seq data.

However, I believe the paper falls significantly short in demonstrating that ZINB-WAVE truly improves on standard workflows. Put another way, from the presented results, I was not excited to try it on datasets from my lab, and indeed when I did try it, I observed essentially no improvement (and in some cases a reduction in signal) in the low-dimensional ZINB-WAVE representation. A few comments below:

1. Figure 2 shows that ZINB-WAVE does a better job of separating cells from different patients, compared to PCA. This is presented as an improvement, but I believe its the opposite! There are shared cell types across all four patients, and ZINB-WAVE should have an improved ability to detect these shared states - not simply separate out the different batches. Perhaps it would be valuable to run ZINB-WAVE using the patient data as a batch covariate, and to see what emerges.

2. The use of silhouette distance as a benchmark is, in my opinion, inappropriate. This is because it is computed in very low-dimensional space ($K=2$ or 3), but these are very complex datasets where 2D representations do not capture the richness of the data. Even so, its clear that the improvements of ZINB-WAVE are minor at best compared to standard PCA, and in some cases the method performs worse.

3. I recognize that it is challenging to identify an optimal benchmarking metric, but the manuscript lacks a clear example from start-to-finish, where ZINB-WAVE can be used to analyze scRNA-seq data and lead to a biological result. For example, what happens if the authors take a publicly available heterogeneous dataset and cluster using ZINB-WAVE distances (or similarly, reconstruct a developmental trajectory), and compare to a standard workflow?

Dear Editor,

Thank you for handling our submission and for the thoughtful and constructive reports. We believe that the revised manuscript has addressed all the issues raised in the reports.

We thank the three reviewers for their positive feedback and constructive comments. Below is a point-by-point response to each report (blue font).

Reviewer #1

I think this is a nice paper that presents a statistically sound and novel method for the analysis of single-cell RNA-seq data. Overall, I like the approach, and I like that the authors have developed a model that truly models the bimodality of the data for dimension reduction. The regression framework is also very relevant for adjusting for biological and technical covariates. In my opinion, it would be a nice addition to Nature Communications. I have several comments/questions listed below, that I think should be useful to the authors to improve their paper and hopefully increase the overall scientific impact.

Major comments:

Biological relevance: While I really appreciate the amount of work that was put into the different comparisons (silhouette scores across methods and normalizations). I would have liked to see an application of ZINB-WAVE that leads to new biological insight, or at least some sort of new application. For example, it would be interesting to see how this method can be used to identify new cell subsets or find novel structure in the data (particularly after batch correction). Or as an alternative, show that some new structures identified in published datasets were likely driven by batch effects that were not accounted for. I am sure there are plenty of those out there!

In retrospect, we agree with the reviewer that to better showcase the capabilities of ZINB-WaVE we need a compelling example of novel biological insights. We therefore present two such examples in the revised version of the manuscript. First, we provide an example of developmental lineage reconstruction, in which we show that the low-rank representation of ZINB-WaVE leads to better pseudotime inference than PCA. Second, we provide a re-analysis of the 10x Genomics 68,000 PBMCs dataset from Zheng et al. (<https://www.nature.com/articles/ncomms14049>) and show that clustering based on the ZINB-WaVE low-dimensional representation is able to identify rare cell-types that were missed by the authors' original analysis (based on PCA). Both analyses are presented in the new Figure 3 of the revised manuscript.

In addition, we provide a more in-depth analysis of the Glioblastoma dataset, showing that the original PCA representation is driven by batch effects and that using ZINB-WaVE with

sample-level covariates we are able to reduce the batch effects without removing the patient differences (revised Figure 5).

Datasets: Most of the dataset used are fairly low throughput. Given the goal of the proposed method (dimension reduction), I think it would be good to try to use a larger dataset with tens of thousands of cells. I suggest the use of the 10X dataset generated by the Bielas lab (<https://www.nature.com/articles/ncomms14049>). It is a large dataset with PBMCs, and sorted cell populations. I think it would be very useful for motivating and validating ZINB-WAVE. This might also help ZINB-WAVE as these data are very sparse with a lot of zero.

As outlined in the response to the previous point, we have added the analysis of the 10x Genomics Chromium 68,000 PBMC's from the suggested article. Our analysis shows that ZINB-WaVE is a feasible approach even with tens of thousands of cells and that our workflow leads to new biological insight.

Comparison to t-SNE: It seems that t-SNE really is the best current practice for dimension reduction and visualization of single-cell data. Given this, I think it would be useful to include t-SNE in your comparison. I am personally not a big fan of t-SNE, and I know that since it's not linear it might not be a fair comparison, but nevertheless I think it would be a useful comparison for the average reader.

We agree with the referee that t-SNE is the most popular method for visualization of single-cell datasets. We do however point out that the inherent stochasticity of the t-SNE projection and the non-trivial interpretation of distances in t-SNE space has led several authors to warn against inferential results based on t-SNE projections. In other words, t-SNE is a great tool for visualization, but great care must be taken when interpreting the structure of the data in t-SNE space. On the other hand, our low-dimensional projection has a straightforward interpretation, since the model in its basic form is a factor analysis model akin to PCA.

The current best practice for using the t-SNE algorithm, and its default implementation, is to first apply some dimensionality reduction technique, typically PCA retaining 50 components, and then apply t-SNE on the Euclidean distance between cells in the reduced PCA space. Following this observation, we explored the possibility of using t-SNE on the Euclidean distance defined by our low-dimensional representation, rather than by PCA. In the revised manuscript, we show how t-SNE can be used to visualize the clustering results of a procedure based on ZINB-WaVE with 10 latent factors: the new Figure 3d in the revised manuscript shows that t-SNE can be used effectively following ZINB-WaVE to give a two-dimensional representation of the 10 latent factors of ZINB-WaVE.

Minor comments:

Although the main goal of the proposed approach is dimension reduction, the zero-inflated model shares some similarities with MAST. ZINB-WAVE is probably better in terms of goodness-of-fit, but it would be nice to show that.

New Supplementary Figure S21 shows goodness-of-fit plots for the MAST model on the V1 dataset. The MAST model fits under-estimate the overall mean $\log_2(\text{TPM}+1)$, over-estimate the zero probability, but its variance estimates are uniformly distributed over the observed proportion of zero counts. We have added a paragraph in the revised manuscript to describe these results.

Number of latent factors. Do you provide any guidance on how many factors should be included in the model? It seems that you only used two for the results presented, is that correct?

The reviewer raises a very important point: the number of factors to be included in the model is indeed a major tuning parameter of our approach and care must be taken to choose a reasonable value. Since our estimation procedure is a maximum likelihood approach, the likelihood function itself can be used to guide the choice of K . In the revised manuscript, we explore the use of the Akaike information criterion (AIC) and Bayesian information criterion (BIC) for the selection of K . We demonstrate with simulations (in revised Supplementary Figure S23) that AIC and BIC are able to identify the right value of K when the model is correctly specified. In the zinbwave software package, we now provide functionality to compute the likelihood of the fitted model as well as AIC and BIC.

Down-selection of genes? Is the proposed approach sensitive to the number of genes used? Why use the 1000 most variables genes, and not the top 500 or 100? How does the approach scale with the number of genes used? How fast is the approach?

The reviewer raises two separate questions: the robustness of the projection to the number of genes used and the computational performance of the method as a function of the number of genes used. To answer the first question, we added a new figure, Supplementary Figure S4, that shows that ZINB-WaVE is rather robust to the selected number of highly variable genes, leading to very similar projections when selecting the 100 through 5,000 most variable genes. To answer the second question, we improved Supplementary Figure S30 to study how ZINB-WaVE's performance scales with the number of genes, cells, and latent factors.

Wanted vs Unwanted variation: Is there a way to know if some of the variation observed in your lower dimensional space is biologically relevant or unwanted (due to nuisance factor). I understand this is difficult to do, but it would be nice to discuss this and perhaps provide some intuitions. Perhaps the first recommendation would be to see if cells cluster according to some technical variable (plate, date, etc). Again, this might be useful to the reader.

We thank the reviewer for raising another important point. We agree with the reviewer that it may be difficult to understand if the observed variation in the low-dimensional data

representation is technical or biological. In our original Figures 2 and S4-S6, we reported the absolute correlation between the low-dimensional factors (estimated by PCA, ZIFA, ZINB-WaVE) and a set of quality control (QC) metrics that can be used to evaluate whether the observed variation is biologically relevant or unwanted. We now realize that this point was lost in the broader message of the figure and we added a paragraph about this in the Discussion.

In the introduction, you say “many genes fail to be detected even though they are expressed”. I think it would be important to say that many genes fail to be detected because they are not expressed, so it might be good to clarify this.

We agree, it is important to underline that genes may be undetected simply because they are not expressed. We have clarified this point in the text.

In the introduction, you say ““bulk” RNA sequencing, which is only capable of measuring gene expression levels averaged over millions of cells”. This was true 10 years ago, but today one can easily do “bulk” RNA-sequencing on small cell subsets with thousands of cells and possibly dozens of cells if one uses an optimized protocol (e.g. SMART-seq). So I suggest that you modify this statement.

We agree with the reviewer that modern protocols allow researchers to measure gene expression in tens or thousands of cells. However, our original point was that single-cell RNA-seq is needed if a researcher is interested in characterizing cell populations, rather than average gene expression in tissues. We have modified the statement to clarify our point.

Reviewer #2 (Remarks to the Author):

This manuscript describes a new method ZINB-WaVE for analyzing single-cell RNA-seq (scRNA-seq) data. scRNA-seq is an emerging technology for measuring the transcriptome of individual cells. It provides a powerful tool to study cell-to-cell heterogeneity. The data from scRNA-seq experiments contain multiple sources of technical biases and noises. Computational and statistical methods play crucial roles in effectively using such data. This manuscript represents a timely study trying to address an important problem in scRNA-seq analysis. It attempts to represent scRNA-seq data using a low-dimensional structure while accounting for dropouts. The ZINB-WaVE method extends the RUV approach developed previously by the same group to zero-inflated negative binomial count data. It allows one to include sample-level or gene-level covariates in the model. Through simulations and real data, the authors demonstrate that ZINB-WaVE performs better than other dimension reduction methods including

PCA and ZIFA. Overall, the manuscript is clearly written, and the proposed method is useful for single-cell RNA-seq data analysis. I do, however, have several questions that I hope the authors can answer.

Major comments:

1. A nice feature of ZINB-WaVE is its ability to handle gene-level covariates such as GC content and gene length. This is also a major difference between ZINB-WaVE and existing methods such as PCA and ZIFA. However, this important feature and its practical value have not been demonstrated anywhere in the manuscript. It will be very useful if the authors can demonstrate the use of this important feature in real applications.

We agree with the reviewer that the ability of including gene-level covariates is a major feature of our ZINB-WaVE model. This feature is used extensively throughout the manuscript, as it allows us to include a gene-level intercept that act as a normalization factor, enabling us to fit the model directly on unnormalized counts. Although additional features, such as gene length and GC-content, can in principle be included in the model, we failed to find an example using public datasets where the inclusion of such gene-level covariates substantially changed the low-dimensional representation of the data.

We do however envision a scenario in which such feature will become important in the future, when large collaborative efforts will require the ability to integrate data from multiple labs, and including GC-content in the model may be beneficial, given its relation to inter-laboratory differences [1]. We added this consideration to the discussion in the revised manuscript.

References:

1. Love, M. I., Hogenesch, J. B. & Irizarry, R. A. Modeling of RNA-seq fragment sequence bias reduces systematic errors in transcript abundance estimation. *Nature Biotechnology* 34, 1287 (2016).

2. In the ZINB-WaVE model, the offset matrices O_{μ} and O_{π} in formulas (4) and (5) are assumed to be known. Each of these matrices contains $n \times J$ entries. That is a large number of parameters. It is unclear to me why these two offset matrices are known and how they are specified?

In developing the ZINB-WaVE approach, we sought to propose the most general possible model, so that it could be reused in the future by us or other researchers without the need of reimplementing it. For this reason, we decided to implement a very general Generalized Linear Model (GLM) framework, which includes features (like the ability of including offsets) which we do not directly use in the present application. Offsets are a standard concept in GLMs (see for instance [1]). A standard example is the use of a log-linear model for proportions, in which one can consider the total number of occurrences as a fixed, *known* offset (see e.g., [1]). In particular, offsets are used in the context of RNA-seq to include normalization scaling factors in differential expression analysis frameworks without the need to transform the data and hence lose the count properties of the data (e.g., in edgeR [2] and DESeq [3]). Although typically the offsets are one per sample (global scaling normalization), one can have gene-specific and

sample-specific offsets (hence an $n \times J$ matrix of offsets) to include, for instance, within-sample GC-content normalization [4]. In these situations, although the offsets are estimated from the data, they are estimated prior to fitting the model. Hence, we do not need to estimate any parameters in relation to offsets.

References:

1. McCullagh, P., Nelder, J.A. Generalized Linear Model (Second Edition). Chapman and Hall (1989).
2. Robinson, M. D., McCarthy, D. J. & Smyth, G. K. edgeR: a Bioconductor package for differential expression analysis of digital gene expression data. *Bioinformatics* 26, 139 (2010).
3. Love, M. I., Huber, W., & Anders, S.. Moderated estimation of fold change and dispersion for RNA-seq data with DESeq2. *Genome biology*, 15(12), 550 (2014).
4. Risso, D., Schwartz, K., Sherlock, G., & Dudoit, S. GC-content normalization for RNA-Seq data. *BMC bioinformatics*, 12(1), 480 (2011).

3. The ZINB-WaVE procedure also assumes that the number of unknown factors (i.e., unknown cell-level covariates) K is given. However, in practice K is unknown. The present manuscript does not provide any procedure to choose K . Although the authors argue that their method is relatively robust to misspecified K , Figure 5 and other related figures show that the choice of K does influence the bias and MSE. Thus, it is important to provide a method that can help users to objectively choose the optimal K .

We thank the reviewer for raising this important point. We agree with both Reviewers 1 and 2 that this is an important aspect of our method that was left unspecified in the previous submission. As outlined in the response to Reviewer 1, we can use likelihood-based statistics to guide the choice of K . In the revised manuscript, we explore the use of the Akaike information criterion (AIC) and Bayesian information criterion (BIC) for the selection of K . We demonstrate with simulations (in revised Supplementary Figure S23) that AIC and BIC are able to identify the right value of K . In the zinbwave software package, we now provide functionality to compute the likelihood of the fitted model as well as AIC and BIC.

4. When comparing ZINB-WaVE with other methods, the data shown in the scatterplots may be misleading. In Figure 2, Figure S4, and Figure S5, the scatterplots for PCA and ZIFA are not based on their best performance. The authors should show the plots from the best normalization method (FQ). The authors showed how the normalization method affects the dimension reduction results of PCA in Figure S8. It will be interesting to see how it affects the results in all 4 real datasets for both PCA and ZIFA.

The reason why we showed the results of PCA and ZIFA based on total-count normalization in the original Figures 2, S4, and S5 is that this is by far the most popular normalization used in the single-cell literature. In fact, total-count normalization was used in the original analysis of all the

datasets used in Figure 4. Moreover, total-count normalization was the method of choice in the ZIFA paper. We therefore argue that this is the most appropriate comparison for the scatterplots of Figure 2.

One of the key point of the manuscript is to show that normalization impacts the results of PCA and ZIFA. Figure 4 provides such a comparison, and allows the reader to compare the performance of our method to that of ZIFA and PCA with their respective best performing normalization. We do agree with the reviewer that showing the scatterplots of PCA and ZIFA following all the normalization methods is interesting and we added new supplementary figures in the revised version of the manuscript to show the effect of normalization on PCA and ZIFA: Specifically, Figure S8 -- S15.

5. The authors showed that their method can be used to adjust for batch effects. To more convincingly demonstrate this advantage, they should compare ZINB-WaVE with PCA and ZIFA in the real data analysis. For example, what if one first uses existing batch effect correction methods such as ComBat to adjust for batch effect and then runs PCA and ZIFA? How does this approach perform compared to ZINB-WaVE?

We agree with the reviewer that comparing the ability of ZINB-WaVE to adjust for batch effects with a procedure based on ComBat and PCA is an interesting analysis that should be added to the paper. To better highlight the similarities and differences of these two approaches, we included an additional analysis of the Glioblastoma data (see revised Fig. 5).

Briefly, the mESC data is an example of good experimental design, where each batch includes cells from each biological condition (this is known as a factorial design). Hence, it is relatively easy to correct for batch effects and and, unsurprisingly, both ComBat and ZINB-WaVE successfully do so (new Supplementary Fig. S16). The advantage of ZINB-WaVE is the ability of including batch effects in the same model used for dimensionality reduction, without the need for prior data normalization.

The Glioblastoma dataset is an example of a more complex situation, in which there is confounding between batch and biology, each patient being processed separately except one, who was processed in two batches. ComBat was not able to correctly account for batch, removing the patient effects along with the batch effects (new Supplementary Fig. S17), while including the detection rate as a covariate in the ZINB-WaVE model led to the removal of the batch effect, while preserving the biological differences between patients (new Fig. 5f).

Overall, we believe that these new analyses show that ZINB-WaVE is more flexible and leads to better batch effect corrections than a two-step procedure involving ComBat and PCA/ZIFA.

6. The authors used the average silhouette width to evaluate the clustering performance of each method. In real applications, the true clustering structure is unknown. Therefore, it will be useful if the authors also evaluate different methods by first performing clustering based on the

low-dimensional signals extracted by different methods and then evaluate how accurate the clustering analysis recovers the true cell clusters (identities).

We thank the reviewer for the good suggestion of an additional comparison to evaluate the different methods. We used the precision and recall coefficients to measure the ability of clustering to correctly recover the true cluster labels in simulated data. The results are presented in the new Supplementary Figure S29 and show how ZINB-WaVE low-dimensional representations lead to better clustering. We added a new paragraph in the Methods section with the details of the comparison.

Minor comments:

7. Page 4, paragraph 2, line 2: “and the second component was highly correlated with dropout and detection rates (Fig. 2d)”. Here “the second component” in Fig 2d does not correlate highly with dropout. The “first” component in Fig 2d does.

We thank the reviewer for finding this typo and we fixed the text in the new version.

8. Page 4, paragraph 3: “These observations are not limited to the Glioblastoma dataset, and the same trend was observed for ... S1/CA1 dataset (Supplementary Fig. S5), ...”. This statement is inconsistent with Supplementary Fig. S5 in which the first dimension of ZINB-WaVE showed higher correlation with coverage and detection rate than ZIFA, and higher detection rate than PCA.

We fixed the text to better reflect the different behaviors of the methods in the S1/CA1 dataset.

9. Page 15, section “Initialization”, step 3: “Set $Z_{ij}=1$ if (i,j) belongs to P ”. Here if (i,j) belongs to P , Z_{ij} should be 0 instead of 1 according to the definition of Z_{ij} .

We thank the reviewer for spotting this error. We fixed the text in the revised version of the manuscript.

Reviewer #3 (Remarks to the Author):

Risso et al. propose a new method, ZinbWave, that extends the RUV model to account for noise models that are specific to scRNA-seq. They test their model on simulated and real data, benchmarking against PCA and ZIFA.

From a statistical standpoint, I enjoyed reading the manuscript. It lays out the problem clearly, and the ZINB-WAVE model does account for the vast majority of (at least known) sources of noise for scRNA-seq. Specific features I appreciated included :

1. The use of both sample and gene-level covariates. Most models do not consider the possibility of gene-level covariates, but this is an interesting addition (however there is no analysis shown on how much these improve the results, which is a shame)
2. Since the model includes size factors, it can run on non-normalized counts. Therefore, the procedure handles normalization and dimensional reduction in a single integrated step, and I believe this is well-motivated and a desired property for scRNA-seq workflows.
3. The optimization and learning procedure is non-trivial, to say the least, though there is always some concern that the procedure could converge to a local minima, which is not a concern for PCA.

Therefore, I believe the manuscript represents a theoretical advance for the analysis of scRNA-seq data.

However, I believe the paper falls significantly short in demonstrating that ZINB-WAVE truly improves on standard workflows. Put another way, from the presented results, I was not excited to try it on datasets from my lab, and indeed when I did try it, I observed essentially no improvement (and in some cases a reduction in signal) in the low-dimensional ZINB-WAVE representation. A few comments below:

We thank the reviewer for recognizing the methodological advances of our paper. We have tried our best in the revised version to provide a more compelling case for the practical advantages of the ZINB-WaVE approach.

1. Figure 2 shows that ZINB-WAVE does a better job of separating cells from different patients, compared to PCA. This is presented as an improvement, but I believe its the opposite! There are shared cell types across all four patients, and ZINB-WAVE should have an improved ability to detect these shared states - not simply separate out the different batches. Perhaps it would be valuable to run ZINB-WAVE using the patient data as a batch covariate, and to see what emerges.

We agree with the reviewer that there is a case to be made that the clustering by patient in the Patel data is an unwanted result. In fact, because only one batch of cells was collected for all but one patient, patient effects are confounded with batch effects, making it impossible to tell whether cells cluster by patient for technical or biological reasons. Moreover, because the patients have different subtypes of glioblastoma, including patient as a batch covariate will also remove most of the biological signal (since each patient contains almost exclusively cells of one subtype, see Figure 4 in Patel et al. (ref. 6 of the revised submission)). However, when we limit our analysis to patient MGH26, which was processed in two batches, we can see that, including detection rate as a covariate, ZINB-WaVE was able to remove the batch effect without removing the patient effect. We have added this analysis of the Glioblastoma dataset in the revised version of the manuscript (new Figure 5).

2. The use of silhouette distance as a benchmark is, in my opinion, inappropriate. This is because it is computed in very low-dimensional space ($K=2$ or 3), but these are very complex datasets where 2D representations do not capture the richness of the data. Even so, it's clear that the improvements of ZINB-WAVE are minor at best compared to standard PCA, and in some cases the method performs worse.

We respectfully disagree with the reviewer, and we believe that average silhouette widths are an effective measure to compare clustering results. This is shown in simulations (Figure 7) in which we know the true labels of the samples. In simulated data, we can show that the silhouette width results (Figure 7c and d) recapitulate the results obtained by computing the correlation between true and estimated distances (Figure 7a and b; a gold-standard measure unavailable in real data).

To further corroborate the silhouette results, we computed precision and recall coefficients, to compare the true cluster labels with the inferred cluster labels obtained by cluster analysis after each dimensionality reduction method (new Supplementary Fig. S27). The results confirmed the good performance of ZINB-WAVE in simulations and that the silhouette width is able to correctly rank the methods.

3. I recognize that it is challenging to identify an optimal benchmarking metric, but the manuscript lacks a clear example from start-to-finish, where ZINB-WAVE can be used to analyze scRNA-seq data and lead to a biological result. For example, what happens if the authors take a publicly available heterogeneous dataset and cluster using ZINB-WAVE distances (or similarly, reconstruct a developmental trajectory), and compare to a standard workflow?

In retrospect, we agree with the reviewer (and with reviewer #1) that the previous version of this manuscript lacked a compelling biological example to show the advantages of our method over typical workflows. The revised version of the paper includes additional analyses of public datasets that hopefully achieve this goal. First, we provide an example of developmental lineage reconstruction, in which we show that the low-rank representation of ZINB-WAVE leads to better pseudotime inference than PCA (the full workflow is presented in <https://f1000research.com/articles/6-1158/>). Second, we provide a re-analysis of the 10x Genomics 68,000 PBMCs dataset from Zheng et al. (<https://www.nature.com/articles/ncomms14049>) and show that clustering based on ZINB-WAVE is able to identify rare cell-types that were missed by the authors' original analysis, based on PCA. Both analyses are presented in the new Figure 3 of the revised manuscript and provide examples of how ZINB-WAVE can be used, in conjunction with clustering and lineage reconstruction algorithms, to extract biological insight from complex scRNA-seq datasets.

REVIEWERS' COMMENTS:

Reviewer #1 (Remarks to the Author):

The authors have done a very good job at revising their paper. I think the new version is great, and the results clearly highlight the strength of the proposed approach. I have no further comments.

Reviewer #2 (Remarks to the Author):

The authors have satisfactorily addressed my previous concerns. I only have a minor suggestion for improving clarity of the presentation.

Minor:

Figure 3a,b needs to be better explained or annotated either in the main manuscript or in the figure legend. It was stated that "the inferred MST only correctly identified the neuronal and microvillous lineages, while it was unable to identify sustentacular cells as a mature cell type." The authors should clearly label in the figure which cell clusters correspond to the neuronal, microvillous, and sustentacular cells. Currently, both Figure 3a and 3b have three branches. It is unclear why the authors say that sustentacular cells were not identified.

CENTRE DE BIO-INFORMATIQUE - CENTRE FOR COMPUTATIONAL BIOLOGY

Paris, December 5, 2017

We thank the reviewers for their comments. We provide below point-by-point responses to their questions :

- Reviewer 1 had no question

- Reviewer 2 has a minor comment : « Figure 3a,b needs to be better explained or annotated either in the main manuscript or in the figure legend. It was stated that "the inferred MST only correctly identified the neuronal and microvillous lineages, while it was unable to identify sustentacular cells as a mature cell type." The authors should clearly label in the figure which cell clusters correspond to the neuronal, microvillous, and sustentacular cells. Currently, both Figure 3a and 3b have three branches. It is unclear why the authors say that sustentacular cells were not identified ».
- ***We added in the main text the abbreviations used in the Figure, and clarified in the caption which cell types should be identified at the three lineage endpoints.***

Best regards,

Jean-Philippe Vert